# Evolutionary pathways to SARS-CoV-2 resistance are opened and closed by epistasis acting on ACE2

**Gianni M. Castiglione**[1], **Lingli Zhou**[1], **Zhenhua Xu**[1], **Zachary Neiman**[1], **Chien-Fu Hung**[2], **Elia J. Duh**[1]*

**1** Department of Ophthalmology, Johns Hopkins University School of Medicine, Baltimore, Maryland, United States of America, **2** Department of Pathology, Johns Hopkins University School of Medicine, Baltimore, Maryland, United States of America

* eduh@jhmi.edu

**Data Availability Statement:** All relevant data are within the paper and its Supporting Information files.

## Abstract

Severe Acute Respiratory Syndrome Coronavirus 2 (SARS-CoV-2) infects a broader range of mammalian species than previously predicted, binding a diversity of angiotensin converting enzyme 2 (ACE2) orthologs despite extensive sequence divergence. Within this sequence degeneracy, we identify a rare sequence combination capable of conferring SARS-CoV-2 resistance. We demonstrate that this sequence was likely unattainable during human evolution due to deleterious effects on ACE2 carboxypeptidase activity, which has vasodilatory and cardioprotective functions in vivo. Across the 25 ACE2 sites implicated in viral binding, we identify 6 amino acid substitutions unique to mouse—one of the only known mammalian species resistant to SARS-CoV-2. Substituting human variants at these positions is sufficient to confer binding of the SARS-CoV-2 S protein to mouse ACE2, facilitating cellular infection. Conversely, substituting mouse variants into either human or dog ACE2 abolishes viral binding, diminishing cellular infection. However, these same substitutions decrease human ACE2 activity by 50% and are predicted as pathogenic, consistent with the extreme rarity of human polymorphisms at these sites. This trade-off can be avoided, however, depending on genetic background; if substituted simultaneously, these same mutations have no deleterious effect on dog ACE2 nor that of the rodent ancestor estimated to exist 70 million years ago. This genetic contingency (epistasis) may have therefore opened the road to resistance for some species, while making humans susceptible to viruses that use these ACE2 surfaces for binding, as does SARS-CoV-2.

## Introduction

Severe Acute Respiratory Syndrome Coronavirus 2 (SARS-CoV-2) is closely related to a virus (RaTG13) isolated from the Chinese horseshoe bat (*Rhinolophus affinis*) [1], with circulation of related viruses in bat populations [2–4] and potential spillover into other species susceptible to infection by SARS-CoV-2–related coronaviruses (e.g., pangolin; *Manis javanica*) [5]. These

**Funding:** This work was supported by research grants from the National Institutes of Health (EY022383 and EY022683; to E.J.D.) and Core Grant P30EY001765, Imaging and Microscopy Core Module. The funders had no role in study design, data collection and analysis, decision to publish, or preparation of the manuscript.

**Competing interests:** The authors have declared that no competing interests exist.

**Abbreviations:** ACE2, angiotensin converting enzyme 2; Ang-II, Angiotensin-II; CmD, clade model D; COVID-19, Coronavirus Disease 2019; hrsACE2, human recombinant soluble ACE2; LRT, likelihood ratio test; MLV, murine leukemia virus; RAAS, renin–angiotensin–aldosterone system; RBD, receptor-binding domain; SARS-CoV-1, Severe Acute Respiratory Syndrome Coronavirus; SARS-CoV-2, Severe Acute Respiratory Syndrome Coronavirus 2; WT, wild-type.

zoonotic origins ultimately led to the evolution of a virus that is highly transmissible among humans [2], causing an unprecedented public health emergency [6,7]. A wide phylogenetic range of mammalian species have been demonstrated to be susceptible to SARS-CoV-2 infection, including nonhuman primates, dogs, cats, ferrets, hamsters, and minks [8–15]. Characterizing the entire host range of SARS-CoV-2 is important for identifying the risks of anthroponosis and zoonosis between humans and other species, which can pose a major health risk by forming novel viral reservoirs where new mutations can evolve [16,17]. A recent example is mink farms, where infection by humans led to the evolution of novel viral strains, which have since reinfected human populations [14]. Attempts to predict the host range of SARS--CoV-2 have greatly underestimated the extent to which SARS-CoV-2 can infect certain non-primate species, especially nonfelid carnivores including mink, dog, and ferret [8–14,18]. These predictive methods depend on comparative sequence analyses of angiotensin converting enzyme 2 (ACE2)—the cellular receptor for SARS-CoV-2—and scoring based on sequence and structural homology to the human ACE2 viral binding interface [18–20]. The difficulty of predicting the SARS-CoV-2 host range through these methods demonstrates that the virus can bind a wide range of ACE2 orthologs despite extensive sequence divergence.

Infection by SARS-CoV-2 is mediated by the binding of the viral S protein receptor-binding domain (RBD) to ACE2 [1,21], displaying a nanomolar affinity higher than that of Severe Acute Respiratory Syndrome Coronavirus (SARS-CoV-1) [22]. Structural analyses have identified that the SARS-CoV-2 RBD targets multiple binding "hotspots" within the ACE2 ectodomain, presenting a diffuse and multifaceted binding strategy [23–25]. Thus, it remains unclear why so few species are resistant to infection, since there is extensive sequence diversity within the viral binding domain of ACE2 [18]. The degeneracy of ACE2 sequence requirements for SARS-CoV-2 binding raises the question as to why sequences capable of blocking viral binding have not evolved more frequently by chance; mice (*Mus musculus*) are one of the only mammalian species known to be resistant to SARS-CoV-2 infection [1,21,26]. This exception is striking, as mice do not have naturally circulating SARS-related viruses that may be expected to drive the evolution of ACE2-mediated viral resistance, as observed during sarbecovirus arms races within bat ACE2 [27,28]. One explanation may be that other physiological constraints exist, potentially limiting this amino acid combination to be functionally tolerable only in mice. Genetic context or starting point can determine which sequence combinations ultimately evolve [29], where intramolecular epistasis, a form of context dependence, enables amino acids to have different functional effects depending on residues at other sites [30]. Epistasis can permit compensatory interactions to evolve following an "original" mutation [31–34], ultimately generating completely different amino acid combinations converging on the same structural/functional "solution" [29,35–38]. Conversely, intramolecular epistasis can limit the number of possible amino acids combinations that will confer a given function [39,40]. The opening and closing of evolutionary trajectories by epistasis and pleiotropy could potentially reconcile the dearth of sequences capable of blocking SARS-CoV-2 binding with the wide diversity of ACE2 sequences.

In mammals, ACE2 evolved to serve an essential reno- and cardioprotective role in vivo, mediated through its regulation of the renin–angiotensin–aldosterone system (RAAS) in conjunction with ACE. The signaling peptide Angiotensin-II (Ang-II) generated by ACE—a major clinical target for hypertension—stimulates vasoconstriction, inflammation, and fibrosis responses through the Ang-II/AT$_1$R axis [41,42]. ACE2 carboxypeptidase activity counteracts these effects through conversion of Ang-II to Ang-(1–7), a peptide that induces vasodilatory, anti-inflammatory, and antifibrotic effects through MAS signaling [42,43]. Loss of ACE2 in mice worsens cardiac dysfunction in obesity, increases diabetic kidney dysfunction, increases mortality rates after myocardial infarction, and can have severe effects on cardiac contractility

[41,43–45]. This protective role of ACE2 is largely mediated through its enzymatic processing of Ang II to Ang-(1–7), where exogenous delivery of either ACE2 or Ang-(1–7) can protect against pathogenic features of multiple cardiovascular and kidney diseases [41,45,46]. Given the myriad protective roles of ACE2 enzymatic activity, it may be expected that ACE2 function is highly conserved across species. However, mice display approximately 50% higher ACE2 activity relative to humans [47] and are insensitive to the vasodilatory effects of Ang-(1–7) [43], suggesting that increased ACE2 activity in rodents may have evolved to serve nonvasodilatory protective functions. However, rodents depend on ACE2-mediated degradation of Ang II to main normal blood pressure and cardiovascular homeostasis [44,48–52]. These observations suggest that major physiological differences between species can drive differences in ACE2 function, as seen in the evolution of other protein systems [34,40,53]. Organismal sensitivity to mutations affecting ACE2 activity may be especially pronounced due to the X-chromosome location of the *ACE2* gene [54], where even heterozygous *ACE2* knockout females (−/x) display increased susceptibility to heart and kidney injury [43].

If natural variation across species has evolved to shape ACE2 function, then the evolution of ACE2 sequences due to differences in physiology may alter the latent capacity for viral receptor usage and susceptibility. We therefore investigated whether SARS-CoV-2 binding to ACE2 could be abolished without disrupting ACE2 enzyme function. To expedite this, we took an evolutionary approach that leveraged the natural sequence variation found in mouse ACE2, which SARS-CoV-2 is unable to bind and gain infection [1,21,26]. Here, we identify a specific combination of mutations unique to rodents which fully abolishes RBD binding when inserted into human and dog ACE2, but which in isolation, significantly decreases ACE2 enzyme activity. These detrimental intermediates would likely severely compromise the cardio- and renoprotective functions of ACE2 activity, explaining why these mutations are rare across mammalian species.

## Results

We investigated ACE2–RBD binding as well as ACE2 enzymatic function across a range of boreoeutherian mammals either susceptible or resistant to SARS-CoV-2 infection [human (*Homo sapiens*; XM_017650263.1), dog (*Canis lupus familiaris*; XM_019746337.1), and pangolin (*M. javanica*; XM_017650263.1) versus mouse (*M. musculus*; XM_017650263.1) and Chinese horseshoe bat (*Rhinolophus sinicus*; XM_019746337.1)] [1,10,12,21]. Using flow cytometry, we found a trend of significantly stronger association of the SARS-CoV-2 RBD with both human and pangolin ACE2 relative to that of SARS-CoV-1, consistent with previous studies [22] (Fig 1A–1D). Notably, SARS-CoV-1 and SARS-CoV-2 RBD association was strongest with human ACE2. We also found evidence that the RBD of both SARS-CoV-1 and SARS-CoV-2 S protein could not bind mouse nor bat (*R. sinicus*) ACE2 (Fig 1B–1D), consistent with previous studies [5,10,12,21,55]. To complement these binding assays, we next characterized the ability of ACE2 orthologs to enable SARS-CoV-2 S pseudovirus entry into cells. We transfected ACE2 orthologs into human cells (HEK293T) and exposed these cells to pseudotyped murine leukemia virus (MLV) particles containing the SARS-CoV-2 S protein. Consistent with our flow cytometry data and with previous studies, this assay largely recapitulated the host range of wild-type (WT) SARS-CoV-2 [21], displaying significant infection of cells expressing human, dog, and pangolin ACE2, but not that of mouse or *R. sinicus* bat (Fig 1E). To investigate whether this variation in ACE2–SARS-CoV-2 S binding is mirrored by functional variation in ACE2 enzyme activity, we measured the carboxypeptidase activity of ACE2 orthologs in vitro using a well-characterized fluorometric biochemical assay [47] (Materials and methods; S1 Fig). Importantly, the Ang-II peptide substrate of ACE2 is conserved among

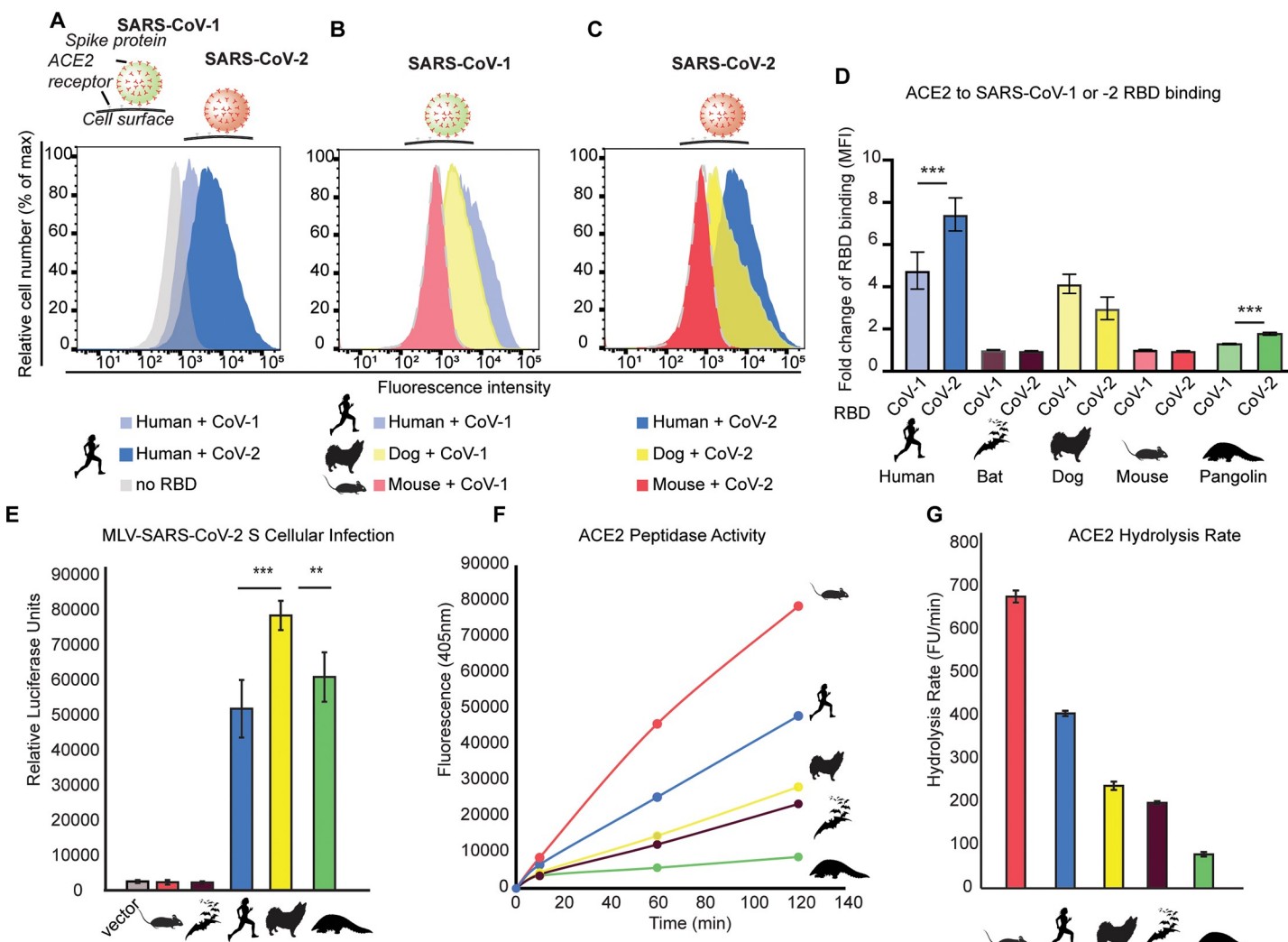

**Fig 1. Natural variation in SARS-CoV-2 binding is mirrored by diversity in ACE2 enzyme activities. (A–D)** Flow cytometry was used to quantify SARS-CoV-1 and SARS-CoV-2 RBD-Fc association with human cells (HEK293T) expressing ACE2–eGFP orthologs from various mammalian species (MFI [22]). $N = 3$ biological replicates. Standard deviation is shown. **(E)** SARS-CoV-2 S pseudovirus infection of HEK293T cells expressing ACE2 orthologs was quantified using a luciferase reporter system. $N = 4$ biological replicates. Standard deviation is shown. **(F)** The carboxypeptidase activity of ACE2 was quantified using a fluorometric peptide incubated with solubilized HEK293T cells transfected with ACE2 orthologs. $N = 4$ biological replicates. **(G)** Hydrolysis rate of ACE2 orthologs (fluorescence units/minute). $N = 3$ to 5 biological replicates. Standard error is shown. All data are available in S1 Data. ACE2, angiotensin converting enzyme 2; MFI, mean fluorescence intensity; RBD, receptor-binding domain; SARS-CoV-1, Severe Acute Respiratory Syndrome Coronavirus; SARS-CoV-2, Severe Acute Respiratory Syndrome Coronavirus 2.

all species investigated here (S2 Fig). Furthermore, all ACE2 protein orthologs displayed highly similar expression levels in HEK293T cells (S3 Fig). Strikingly, we found significant variation in ACE2 hydrolysis rates across all mammalian species, suggesting tuning of enzymatic function across evolutionary history (Fig 1F and 1G), perhaps in response to interspecies RAAS variation. Interesting, dog, *R. sinicus* bat, and pangolin all displayed low ACE2 hydrolysis relative to human and mouse. Mice displayed the highest ACE2 hydrolysis rates, consistent with previous reports [47]. This demonstrates that ACE2 function varies considerably across species and suggests that ACE2 sequence variation could potentially reflect diversification of ACE2 enzymatic function.

To test this, we first took a bioinformatic approach. We reasoned that since ACE2 has evolved under pressures related to its enzymatic processing of Ang-II/Ang-(1–7), signatures of natural selection at sites within the viral binding interface may reflect a potential role of those sites in mediating ACE2 catalytic activity [56]. We therefore searched for shifts in *ACE2* mutational rates ($d_N/d_S$) beyond what may be expected from neutral evolutionary pressures alone [57,58]. To do this, we constructed a phylogenetic dataset representing full-length mammalian *ACE2* sequences encompassing residues 22 to 742 of human ACE2 (S4 Fig, S1 Table). This dataset represents all major mammalian lineages [59], with a high sampling of bats in particular (Chioptera) since they are known to display high ACE2 sequence diversity [27,28]. We conducted a codon-based statistical phylogenetic analysis on the entire tree (PAML, HyPhy) and found significant evidence that mammalian ACE2 is under positive selection, outperforming models assuming neutral molecular evolution (S2 Table). Since synonymous variation can confound tests of positive selection, we ran a test that explicitly accounts for it in model parameters [60]. This analysis also detected significant evidence of positive selection across *ACE2* (S3 Table). As a further control, we excluded bat ACE2 from the analysis, since they are known to display positive selection reflecting an arms race with sarbecovirus binding and infection [27,28]. Even when bats were excluded from the analysis, we still detected evidence of positive selection in mammalian ACE2 (S4 Table). This provides evidence that mammalian ACE2 function may be a target of natural selection that shifts in response to physiological constraints unrelated to sarbecovirus arms races.

To focus our analysis on ACE2 sites responsible for mediating SARS-CoV-2 binding, we took an evolutionary approach leveraging natural sequence variation found in mouse ACE2, which SARS-CoV-2 is unable to bind [1,21]. We identified a set of 6 residues unique to mouse ACE2 that were of particular interest due to their proximity to viral RBD residues implicated in the ACE2–RBD structure (Fig 2A). These sites contact opposite ends of the viral RBD, facilitating binding through a combination of hydrogen bonds (Q24 and K353), as well as van der Waals forces within a hydrophobic pocket of ACE2 (L79, M82, Y83, and P84) [23,25]. These ACE2 residues are located well outside the ACE2 active site mediating catalysis of Ang-II/Ang-(1–7) (Fig 2A; yellow), suggesting that any role of these sites in mediating ACE2 function could be through indirect effects modulating the protein structure, as seen in other protein systems [56,61]. Phylogenetic analysis of evolutionary rates across the ACE2 gene with or without the inclusion of bat ACE2 sequences (S2 and S4 Tables, respectively) identified several of these sites deviating significantly from neutral expectations (Fig 2C; sites 24, 79, dN/dS>1; site 353, dN/dS<1; M8, Bayes empirical Bayes; S5 Table). This finding held even when synonymous rate variation was incorporated into the model (FUBAR; Fig 2C). Interestingly, these sites are less variable in primates relative to other mammalian lineages [19], displaying significantly decreased *ACE2* evolutionary rates relative to bats (Chiroptera) and rodents (Fig 2D, S6 Table) (CmD; dN/dS). Interestingly, 2 of these sites (24 and 79) were previously identified as being under positive selection in bats, likely due to evolutionary arms race between bat ACE2 and sarbecovirus binding and infection [27,28]. The fact that these sites remain under positive selection even after bats are excluded from the analysis (S5 Table) strongly suggests that these sites are evolving in response to physiological variables directly related to ACE2 function, rather than viral binding alone.

Although these bioinformatic signatures suggest that these sites are functionally important, these predictions could have been influenced by the presence of multinucleotide substitutions among ACE2 orthologs [62]. Thus, we experimentally tested whether ACE2 sequence variation at these sites plays a role in the diversification of ACE2 function. We also investigated whether these sites mediate viral binding and infection. First, we substituted the rare residues found in mouse ACE2 with the residues found in human ACE2 (Fig 2B). Using flow

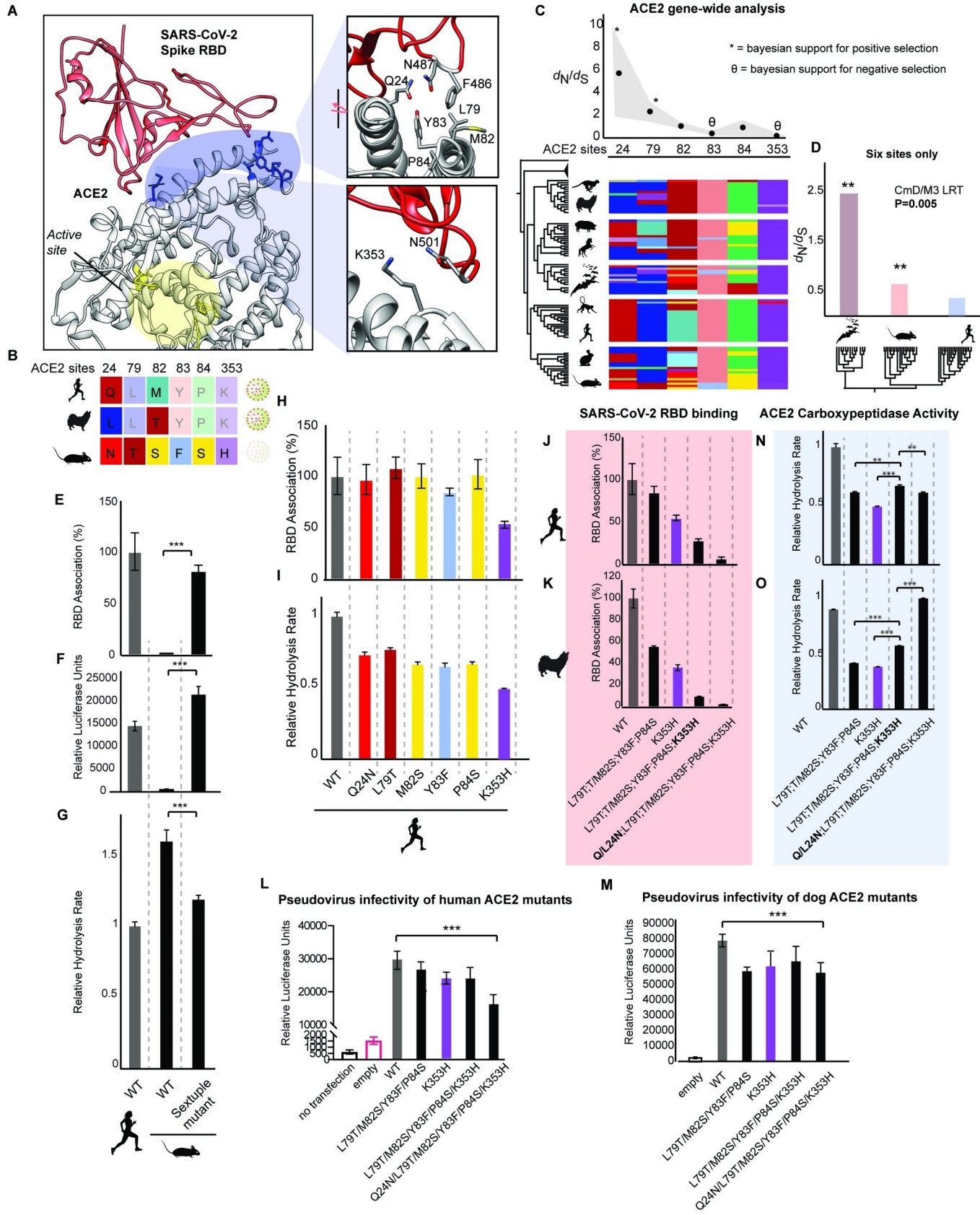

**Fig 2. Interacting amino acids mediating ACE2 activity also govern SARS-CoV-2 binding and cellular infection. (A)** SARS-CoV-2 gains cellular entry through the viral spike protein RBD (red), which targets binding hotspots on the ACE2 receptor (blue) distal to the ACE2 active site (yellow) [6M17; [25]]. **(B)** Mouse ACE2 displays unique amino acid residues at positions within the RBD binding interface relative to other mammals (S1 Table). **(C)** Gene-wide statistical phylogenetic analyses (dN/dS averages [dots] and ranges [gray]; PAML, HyPhy) of an alignment of mammalian ACE2 coding sequences (residues 22 to 742, human ACE2 numbering; uniport ID Q9BYF1) reveals positive (*) and negative selection (θ) on RBD binding hotspots. Alignment of ACE2 residues across boreoeutherian mammals is shown. **(D)** Evolutionary rates of the 6 ACE2 sites that display rare variants in mice. **(E)** Flow cytometry was used to quantify RBD association with human cells (HEK293T) expressing WT human and mouse ACE2, as well as mutant mouse ACE2 containing all 6 human substitutions. $N = 3$ biological replicates; standard deviation is shown. **(F)** Infection of HEK293T cells transfected with either WT or mutant ACE2 exposed to VSV-G pseudotyped with SARS-CoV-2 S protein. Cellular infection was measured as a function of luciferase luminescence. $N = 4$ biological replicates. Standard error is shown. **(G)** The effect of ACE2 mutations on ACE2 hydrolysis rates was measured using a fluorometric peptide. ACE2 activity was measured as fluorescence units per minute. $N = 5$ biological replicates. Standard error is shown. **(H)** Flow cytometry analysis of RBD association with human cells (HEK293T) expressing WT and mutant human ACE2. $N = 3$ biological replicates. Standard error is shown. **(I)** The effect of human ACE2 mutations on ACE2 hydrolysis rates. $N = 3$ to 5 biological replicates. Standard error is shown. **(J, K)** Flow cytometry analysis of RBD association with human cells (HEK293T) expressing WT and mutant human and dog ACE2. $N = 2$ to 3 biological replicates. Standard error is shown. **(L, M)** Pseudovirus infection of HEK293T cells transfected with either (L) human or (M) dog ACE2, containing the indicated mutations. $N = 4$ biological replicates. Standard deviation is shown. **(N, O)** ACE2 hydrolysis activity of dog ACE2 with indicated mutations. $N = 3$ to 5 biological replicates. Standard error is shown. All data are available in S1 Data. ACE2, angiotensin converting enzyme 2; LRT, likelihood ratio test; RBD, receptor-binding domain; SARS-CoV-2, Severe Acute Respiratory Syndrome Coronavirus 2; WT, wild-type.

cytometry, we found that these mutations were sufficient to confer mouse ACE2 with binding to SARS-CoV-2 S RBD, at nearly 80% of that of human WT ACE2 (Fig 2E). Using the MLV-SARS-CoV-2 pseudovirus system, we found that mutant mouse ACE2 containing these 6 substitutions was sufficient to confer pseudovirus infection of HEK293T cells, even beyond that conferred by WT human ACE2 (Fig 2F). This demonstrates that these rare mouse mutations likely play a key role in the resistance of mice to SARS-CoV-2 [1,21]. Interestingly, we find that these 6 mutations altogether also significantly decreased the catalytic activity of mouse ACE2 by over 25% (Fig 2G), suggesting that at least some of these sites are functionally important. Next, we investigated whether these sites also mediate the enzymatic activity and the binding of SARS-CoV-1/2 S RBD to human ACE2. Using co-immunoprecipitation and flow cytometry, we systematically characterized the effect of these rare variants from mouse on the binding of SARS-CoV-1/2 S RBD to human ACE2. The single mutation with the largest effect on the RBD binding of both SARS-CoV-1 (S5 Fig) and SARS-CoV-2 to ACE2 was K353H (Fig 2H). The other single mutations had only minor effects on SARS-CoV-2 RBD binding (Fig 2H). However, each of the 6 single mutations displayed considerable effects on human ACE2 hydrolysis activity (Fig 2I). This demonstrates that ACE2 residues within the RBD binding interface can indirectly modulate ACE2 activity.

Next, we attempted to abolish RBD binding to both human and dog ACE2 by making multiple substitutions at these ACE2 sites. Consistent with the reciprocal experiment in mice described above (Fig 2E), RBD binding to the ACE2 of both human and dog was abolished with just these 6 mouse substitutions (Fig 2J and 2K). This is surprising given the numerous other sites implicated in mediating human and dog ACE2 interactions with the viral RBD [23,25,63]. In both human and dog ACE2, abolition of RBD binding depended on mutating sites 24 and 82 (Fig 2J and 2K), despite displaying different residues in human (Q24; M82) and dog (L24; T82) ACE2 (Fig 2C). This demonstrates that SARS-CoV-2 utilizes the same combination of ACE2 positions to bind both human and dog ACE2 despite amino acid variation at these sites. This may explain why bioinformatic predictions of SARS-CoV-2 host range based on human ACE2 sequence homology have tended to underestimate the infection risk of species such as dogs, ferrets, and minks [9,10,12–14,18]. Using the MLV-SARS-CoV-2 pseudovirus system, we found that these 6 mouse substitutions also significantly reduced pseudovirus infection of HEK293T cells expressing WT and sextuple mutant human and dog ACE2 (Fig 2L and 2M). Although infection was not completely abolished in mutant human and dog ACE2, the reciprocal human substitutions into mouse ACE2 were sufficient to confer infection (Fig

2E). This altogether demonstrates that these rare mouse mutations are likely sufficient for resistance to SARS-CoV-2.

We had observed that these rare mouse substitutions are partly responsible for the high enzymatic activity of mouse ACE2 relative to that of human (Fig 2G). Yet, when substituted into human ACE2, single mutations drive down ACE2 activity (Fig 2I). The dependence of mutational effects on genetic background implies epistasis in ACE2 function. Consistent with this, we observed in both human and dog ACE2 a context dependence (epistasis) of mutational effects between the distal domains of the RBD binding interface. Specifically, we found that the detrimental functional effects of the quadruple mutant in both human and dog ACE2 (L79T; T/M82S; Y83F; P84S) as well as the K353H single mutant were significantly reversed when combined—a phenomenon known as sign epistasis—displaying partial compensation for each other's detrimental effects on ACE2 hydrolysis rates (Fig 2N and 2O). Further evidence of epistasis in this ACE2 domain is seen in the sextuple mutant, where introduction of L24N in dog ACE2 fully rescued ACE2 activity, whereas Q24N decreased activity in human ACE2 relative to the quintuple mutant (Fig 2N and 2O). This discrepancy is likely attributable to amino acid interactions between site 24 and other sites not investigated. These results demonstrate that SARS-CoV-2 binding and infection depend on functionally important ACE2 sites that are highly sensitive to background effects (intramolecular epistasis).

Since normal ACE2 activity is essential for blood pressure regulation and cardiac homeostasis (Fig 3A and 3B; [43]), the deleterious functional effects of these mutations may have closed this human ACE2 evolutionary trajectory leading to SARS-CoV-2 resistance [31,40]. Indeed, we found extremely low human allele frequencies for missense polymorphisms at these ACE2 sites (gnomAD), with most at zero (Fig 3C). It is notable that the 2 sites with polymorphisms (82 and 84) are the only 2/6 where we detected no evidence of positive or negative selection (Fig 2C). We investigated the predicted pathogenicity of these polymorphisms and compared them to the hypothetical polymorphism K353H (Fig 3D)—a substitution required to abolish SARS-CoV-2 binding at the cost of large detrimental effects on human ACE2 activity. We employed PolyPhen—a machine learning–based Bayesian method for estimating pathogenicity of nonsynonymous human mutations ([64]). Consistent with its relatively high allele frequency ($2.44 \times 10^{-5}$), the M82I polymorphism had a zero PolyPhen score, indicating it as benign (Fig 3D). The nearly 5-fold less common polymorphism P84T ($5.47 \times 10^{-6}$) had a slightly higher pathogenicity score but was still predicted as benign (Fig 3D). By comparison, the hypothetical human mutation (K353H) produced a "possibly damaging" PolyPhen score (Fig 3D), consistent with the 2-fold decrease in human ACE2 activity caused by the K353H mutation (Fig 2H), the absence of human polymorphisms at this site (Fig 3C), and strong purifying selection across mammals (Fig 2C). Interestingly, a human ACE2 polymorphism directly adjacent to site 353 (G352V) produced a "probably damaging" PolyPhen score (Fig 3D), consistent with its low allele frequency ($5.75 \times 10^{-6}$). This analysis suggests that polymorphisms at sites 352 to 353 may be unlikely to evolve in human populations due to functionally deleterious effects. This functional constraint likely blocks evolutionary escape of human ACE2 from SARS-CoV-2 binding, which depends on K353H (Fig 3E). In dog ACE2, many single and combinatorial mutations at these sites are also functionally detrimental, including K353H (Fig 3F). Unlike human ACE2, however, in the dog ACE2 background, the sign epistasis of K353H induces fully compensatory effects, such that simultaneously substituting all 6 mutations becomes functionally nearly neutral (Fig 3F, dashed yellow arrow). This suggests that unlike humans and other primates, evolutionary escape from CoV-2 binding may be possible in canines along future evolutionary trajectories.

We hypothesized that the situation may have been analogous in the rodent ancestor 70 MYA; these 6 mutations may have represented a viable alternative sequence–function

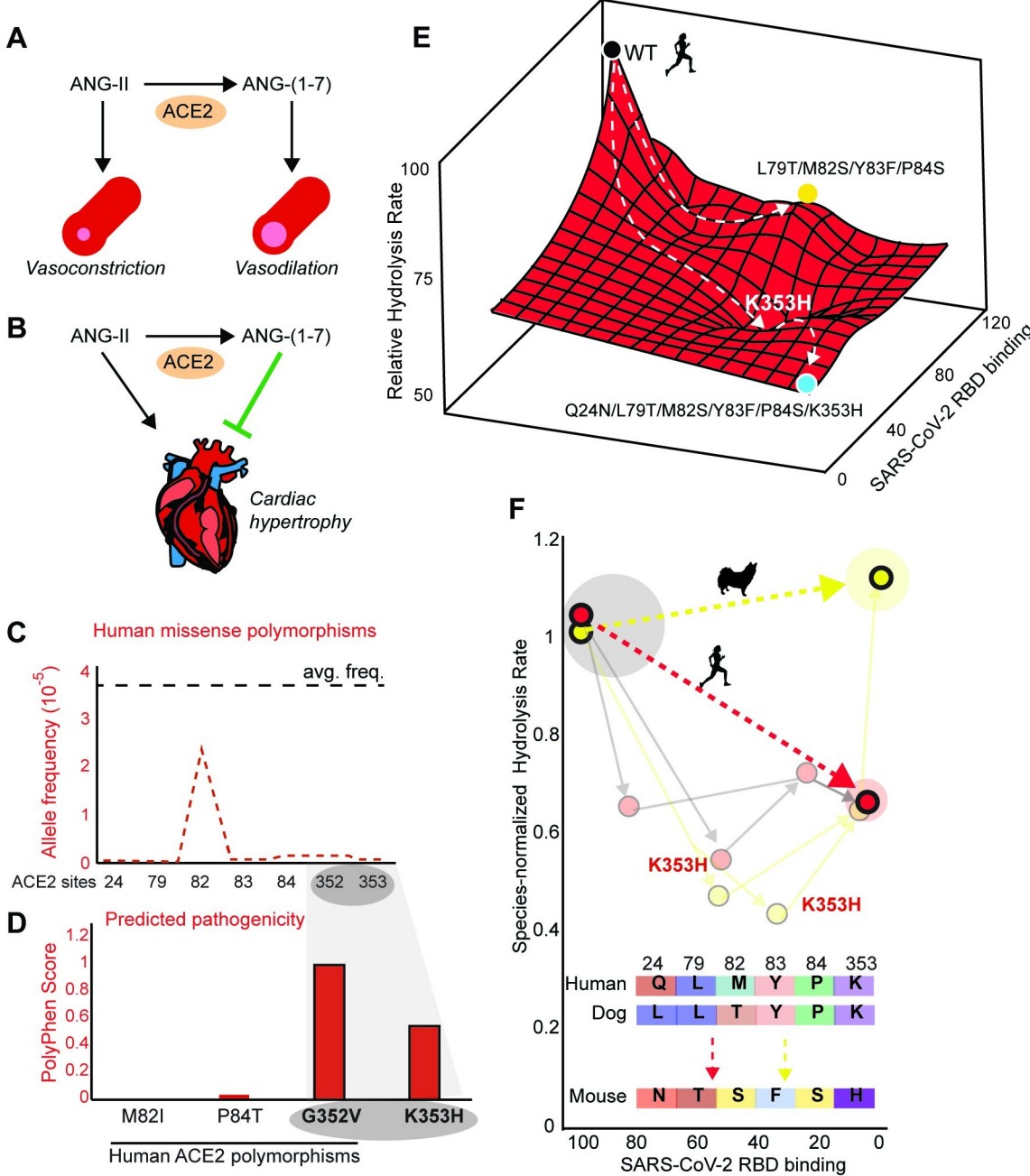

**Fig 3. Human polymorphisms sufficient for SARS-CoV-2 resistance are blocked by deleterious functional effects related to cardiovascular constraints. (A, B)** Hydrolysis of ANG-II by ACE2 generates ANG-(1–7), a vasodilatory peptide with protective effects against cardiac hypertrophy [43]. **(C)** Allele frequencies for human missense polymorphisms at these positions are lower than the protein-wide average frequency at a given ACE2 site (black line). Site 352 is included for comparison to site 353, which had no polymorphisms. **(D)** Predicted pathogenicity of human ACE2 polymorphisms and the hypothetical substitution (K353H) sufficient for resistance to SARS-CoV-2 (PolyPhen). **(E)** Substitution of mouse residues into human ACE2 abolishes ACE2–RBD binding, but with large trade-offs on ACE2 hydrolytic activity due to K353H. **(F)** Unlike human ACE2, dog ACE2 can theoretically reach an alternative sequence that abolishes viral binding without deleterious trade-offs on ACE2 activity. For comparison, each species ACE2 hydrolysis rates are normalized to respective WT values. All data are available in S1 Data. ACE2, angiotensin converting enzyme 2; RBD, receptor-binding domain; SARS-CoV-2, Severe Acute Respiratory Syndrome Coronavirus 2; WT, wild-type.

optimum, thus facilitating their eventual evolution in the *mus* genus. To test this, we constructed a large jawed vertebrate *ACE2* phylogenetic dataset (Fig 4A, S6 Fig) and reconstructed the ancestral rodent (RodAnc) *ACE2* using likelihood methods [65]. Since the carboxyl terminus of vertebrate *ACE2* was too divergent to align, we constructed the RodAnc ACE2 coding-sequence with either mouse (-M) or human (-H) carboxyl termini and measured ACE2-specific hydrolysis activity. We found that RodAnc-M displayed high ACE2 hydrolysis activity indistinguishable from that of mouse (Fig 4B). Moreover, the RodAnc with the human carboxyl terminus displayed ACE2 activity significantly lower than mouse ACE2, but significantly higher than that of human, suggesting that the carboxyl terminus is necessary, but not sufficient to explain the high ACE2 activity seen in the RodAnc. We therefore continued our mutagenesis investigation using RodAnc-M (hereafter referred to as RodAnc). RodAnc ACE2 had none of the rare sequence variants unique to the RBD binding domain of mouse ACE2 (Fig 4C). This strongly suggests that these mutations conferring mice with resistance to SARS-CoV-2 infection appeared relatively recently in rodent ACE2 evolution. Consistent with this, RodAnc ACE2 bound SARS-CoV-2 S RBD at a level nearly identical to that of human ACE2, with viral binding abolished after introduction of the 6 mouse mutations (Fig 4D). Consistent with our hypothesis, we found that RodAnc could tolerate the introduction of all 6 mouse mutations without deleterious effects on ACE2 enzymatic activity (Fig 4E). The functional viability of this alternative sequence combination in the genetic background of RodAnc *ACE2* may therefore explain why these rare variants were ultimately permitted to evolve in mice.

If the functional viability of this alternative sequence explains the traversal of this trajectory by mice, then why was this alternative sequence combination never realized in dog ACE2, where it is, a priori, equally viable? We hypothesized that mammalian differences in physiological constraints related to ACE2 function may influence these evolutionary trajectories. Although ACE2 activity plays a critical role in cardioprotection and prevention of high blood pressure in both mice and humans [43,66], a key physiological difference in rodents is small body size, which is known to result in a lower homeostatic blood pressure set point relative to larger body size mammals [67]. We conducted a phylogenetic statistical analysis on systolic blood pressure values across mammalian species and observed a significant correlation with body size ($r^2$ = 0.26; $p$ = 0.001; phylogenetic independent contrast least squares linear regression) (Figs 4G and S7). Blood pressures were lowest in rodent and bat species and much higher in carnivores, including dog, as well as primates (Fig 4G). Since ACE2 is critical for degrading ANG-II and preventing ANG-II-mediated vasoconstriction in rodents [44,48–52], we propose that high ACE2 activity may have evolved in the rodent ancestor to maintain the lower homeostatic blood pressure set point in rodents relative to large mammals, such as primates and carnivores. This is consistent with fossorial data suggesting that the RodAnc was among the smallest ancestral mammal 70 MYA [68]. Although preliminary, these observations suggest that low blood pressure, and by extension, high ACE2 activity, may be a prerequisite to traverse this evolutionary trajectory, even if it ultimately leads to a sequence conferring equivalent activity (Fig 4F). By contrast, high ACE2 activity may be unlikely to evolve in carnivores and primates due to blood pressure constraints, therefore closing this trajectory. Although speculative, we discuss below how these genetic and physiological contingencies may have indirectly influenced the evolution of resistance to SARS-CoV-2.

## Discussion

We have shown that the ACE2 evolutionary trajectory leading to SARS-CoV-2 resistance in mice [26] was likely unattainable during human evolution. This is caused by the fact that the 6 mutations abolishing SARS-CoV-2 S protein RBD–ACE2 binding also individually disrupt

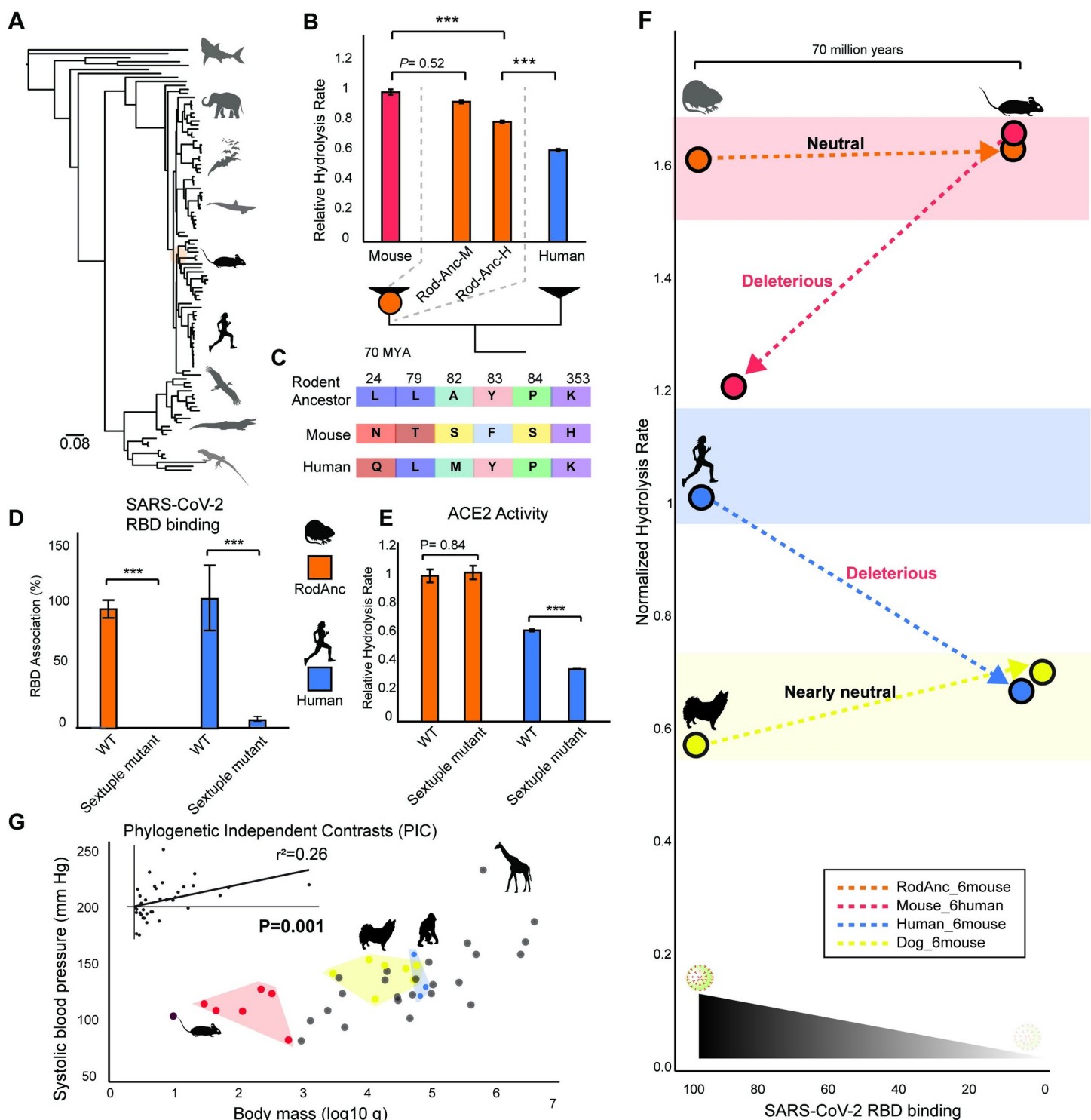

**Fig 4. Alternative sequence–function optima in rodent ancestor ACE2 serendipitously confers resistance to SARS-CoV-2 in extant mice. (A)** Phylogeny of jawed vertebrates used in the ancestral reconstruction of *ACE2* from the last common ancestor of rodents (RodAnc). **(B)** Evolution of high ACE2 activity in the rodent ancestor. The carboxyl terminus of mouse (M) or human (H) was used to determine hydrolysis rates of a fluorometric peptide. *N* = 4 to 5 biological replicates. Standard error is shown. **(C)** RodAnc ACE2 displays none of the rare sequence variants unique to the RBD binding domain of mice. **(D)** Flow cytometry analysis of SARS-CoV-2 S RBD binding to RodAnc-M and human ACE2. Sextuple mutants of each contain the 6 mouse variants. *N* = 3 biological replicates. Standard deviation is shown. **(E)** Hydrolysis rates of WT and mutant ACE2. *N* = 5 biological replicates. Standard error is shown. **(F)** Residues conferring mice with resistance to SARS-CoV-2 are unlikely to evolve in humans due to antagonistic pleiotropy with human ACE2 hydrolysis rates. This antagonistic pleiotropy is absent from dogs and the rodent ancestor. All data points

relative to human WT, except dog, for which RBD binding is shown relative to dog WT. **(G)** Systolic blood pressure divergence between rodents (red) and primates (blue) is correlated with differences in body size ($p = 0.001$; phylogenetically independent least squares linear regression). All data are available in S1 Data. ACE2, angiotensin converting enzyme 2; PIC, phylogenetic independent contrast; RBD, receptor-binding domain; SARS-CoV-2, Severe Acute Respiratory Syndrome Coronavirus 2; WT, wild-type.

ACE2 hydrolysis activity by up to 50% to 60%. These ACE2 mutations are fully compensatory in combination, yet only in the genetic background of dog and the rodent ancestor, not human. Although it is possible that these sites directly modulate substrate binding, it is known that angiotensin substrates bind within a deep cleft housing the active site [69], whereas RBD binding occurs on the outer surface of ACE2 [25]. Interestingly, RBD binding is known to increase ACE2 activity [70], suggesting that RBD binding does not impede substrate recognition. Our results therefore suggest that like other proteins, the catalytic activity of ACE2 can be mediated by the indirect structural effects of residues outside the active site, which may be modulated by RBD binding [40,56,61]. Although previous work demonstrated that inactivation of the ACE2 active site had no effect on SARS-CoV-1 and SARS-CoV-2 binding, these mutations were directly within the active site far from the viral binding interface [71,72]. The mutations investigated here are directly within the viral binding interface of SARS-CoV-2 and from species (*M. musculus*) naturally resistant to SARS-CoV-2 that still maintain ACE2 activity. While it is conceivable that human ACE2 could perhaps have evolved reduced viral binding affinity through other mutations that do not affect enzymatic activity, a previous study showed that human ACE2 engineered to increase viral binding (T27Y, L79T, and N330Y) also displays reduced ACE2 activity [73]. Here, we discuss the implications of epistasis and antagonistic pleiotropy for the evolution of ACE2 function and potential impacts on therapeutic design.

Our work demonstrates the essential importance of the ACE2 viral binding domain on catalytic function. The consequent constraints on the ACE2 amino acid sequence in humans and other primates strongly indicate that protective polymorphisms in the human population are unlikely, in contrast to examples such as CCR5, the co-receptor for HIV [74]. Our analysis shows that allele frequencies for ACE2 missense polymorphisms at 6 sites necessary for SARS-CoV-2 binding are lower than the protein-wide average, reflecting the physiological importance of these positions in mediating ACE2 catalytic activity. Although potentially protective ACE2 polymorphisms have been predicted at other sites in the RBD–ACE2 interface [75,76], we show that statistically unlikely combinations of mutations are sufficient to disrupt SARS-CoV-2 binding, each of which have severe effects on ACE2 activity. Given the importance of ACE2 enzymatic processing of Ang II to Ang-(1–7) in protection against pathogenic features of multiple cardiovascular and kidney diseases [41,45,46], it is possible that that the extant mutational combinations observed in nature (e.g., human, bat, dog, mouse, and pangolin) may all represent alternative sequence "solutions" [29,77] each uniquely required for an animal's physiology, thereby representing fitness "peaks" in the sequence landscape [29,31,34,78–81]. Alternatively, the lack of evolutionary conservation at these functionally relevant sites may suggest that mutational effects on ACE2 activity are not physiological relevant. Other RAAS components may be more important, such as renin activity levels, which are a major factor in determining circulating Ang-II levels in humans [43,82,83]. Yet, there is a known association of human ACE2 polymorphisms with hypertension [84,85], and our analysis suggests that at least 1 of the 6 mutations sufficient for resistance to SARS-CoV-2 may be pathogenic. Selective processes may have therefore partly influence ACE2 evolution, which was likely complicated by the effects of epistasis and pleiotropy, as discussed below.

The effect of mutations on protein function is often complicated by pleiotropy and epistasis [31,86]. Both phenomena can constrain evolutionary trajectories by forcing a dependence on background genotype and phenotype [31,34,40,81,87–90]. Here, we find that these 2 phenomena interact in a complex way to block humans from a trajectory that led to SARS-CoV-2 resistance in mice. Specifically, we find that sign epistasis determines the occurrence of antagonistic pleiotropy: In dog ACE2, all 6 mouse mutations are fully compensatory, whereas in humans, no such alternative sequence–function optima exist. In the rodent ancestor ACE2, the alternative optima also existed as it currently does in dog, suggesting that dog ACE2 could eventually converge on this sequence combination. However, evolution can take different pathways depending on the genetic starting point [29], and ostensibly accessible alternative optima can be blocked by environmental and physiological constraints [34,91,92]. The occurrence of this sequence combination only a few times in evolutionary history despite widespread sequence degeneracy suggests that constraints exist, which limit the evolvability of this sequence. For instance, the existence of functionally detrimental intermediates may explain why these sequences did not evolve more frequently in evolutionary history [29,31,34,78–80]. Notably, the fitness effects of pleiotropy can change with genetic and environmental background [87,88], suggesting the detrimental intermediates may have had more minimal fitness effects in the rodent ancestor, potentially due to higher basal levels of ACE2 activity in rodents. Although preliminary, our evidence suggests that unique cardiovascular constraints in rodents may have opened the available pool of sequence variation, allowing sequence diversification in the rodent ancestor.

Our study also has relevance for therapeutic design. Human recombinant soluble ACE2 (hrsACE2) is in active development as a strategy to neutralize SARS-CoV-2 by binding the viral spike protein [83,93]. In human Coronavirus Disease 2019 (COVID-19) patients, the catalytic activity of hrsACE2 can help reduce angiotensin II levels as well as inflammation associated with COVID-19, likely through elevating Ang-(1–7) levels [83]. hrsACE2 may therefore have the added benefit of minimizing the injury to multiple organs caused by viral-induced downregulation of ACE2 expression and renin–angiotensin hyperactivation [44,94–98]. However, under normal physiological conditions, higher sACE2 plasma activity has been associated with increased pulmonary artery systolic pressure and ventricular systolic dysfunction [99]. In these instances where patients have preexisting conditions that can be exacerbated by increasing sACE2 activity, it may be beneficial to make use of catalytically inactive hrsACE2 that binds RBD with similar efficiency [71]. There is likely to be a spectrum of scenarios where variable levels of ACE2 activity is desirable, in which case recombinant nonhuman ACE2 can serve a key role.

It is important to note the caveats of our interpretations. In addition to its main role in Ang-II processing, ACE2 carboxypeptidase activity can also process other peptide mediators of RAAS signaling, including Ang-I and Angiotensin A [43]. ACE2 also plays an important role in the intestine as a trafficking adaptor for the large amino acid transporter B(O)AT1, which is essential for regulating tryptophan levels in blood [100,101]. This suggests that natural variation in mammalian ACE2 may represent adaptation to a diversity of physiological processes mediated by these important ACE2 functions. Second, our experimental characterizations have limited ACE2 sampling. For instance, we did not investigate other bat ACE2 orthologs, which vary in their ability to facilitate SARS-CoV-2 infection [102] and may have other receptors permitting entry. Similarly, we did not investigate other ancestral intermediates between the rodent ancestor and mouse, which may reveal additional mutational combinations at these sites with varying degrees of epistasis and pleiotropy. Last, we only explored a subset of sequence–function space in human ACE2, raising the possibility that other evolutionary pathways to SARS-CoV-2 resistance may exist without deleterious trade-offs on ACE2

function. Even considering these caveats, our results provide evidence that ACE2 likely evolves in response to functional constraints, which limit the accessibility of evolutionary trajectories, one of which may have led to resistance to SARS-CoV-2.

## Materials and methods

### ACE2 and SARS-CoV-2 constructs

WT hACE2 with a 1D4 (C9) carboxyl-terminal tag (TETSQVAPA) was a gift from Hyeryun Choe (Addgene plasmid # 1786; http://n2t.net/addgene:1786; RRID:Addgene_1786) [103]. Site-directed mutagenesis primers were designed to induce single amino acid substitutions via PCR (QuickChange II, Agilent, Santa Clara, California, USA). Mouse ACE2 was cloned from cDNA synthesized from an RNA extraction of ileum tissue from a C57/BL6 mouse sacrificed in compliance with all regulations of the Johns Hopkins University Institutional Animal Care and Use committee. ACE2 coding sequences from dog (XM_014111329.2), pangolin (*M. javanica*; XM_017650263.1), bat (*R. sinicus*; XM_019746337.1), and the rodent ancestor were synthesized as gblocks (Integrated DNA Technologies, Coralville, Iowa, USA). All animal ACE2 inserts were cloned into a pEGFP-N1 vector, with eGFP carboxyl-terminal tag. For interspecies comparisons, hACE2 was also cloned into pEGFP-N1. The RBD of the SARS-CoV-2 S protein was obtained as a pcDNA3-SARS-CoV-2-S-RBD-Fc plasmid, as a gift from Erik Procko [73] (Addgene plasmid # 141183; http://n2t.net/addgene:141183; RRID: Addgene_141183). pcDNA3.1-SARS-Spike was a gift from Fang Li (Addgene plasmid # 145031; http://n2t.net/addgene:145031; RRID:Addgene_145031) [23]. We used this plasmid as a template to clone the RBD of SARS-CoV-1 Spike protein into pcDNA3 with a Fc carboxyl-terminal tag. This RBD consisted of residues 318 to 510, as previously described (numbering as per AAP13441.1) [104].

### Flow cytometry and immunoprecipitation

HEK293T cells were transfected with ACE2-1D4, or SARS-CoV-2 S protein RBD-Fc constructs using TransIT-X2 (Mirus, Madison, Wisconsin, USA). Twenty-four hours after transfection, the media of RBD-transfected cells was replaced with OptiPRO SFM (Thermo Fisher, Waltham, Massachusetts, USA). Seventy-two hours after transfection, the media from RBD-transfected cells was concentrated using an Amicon Ultra-15 centrifugal filter unit with a 3,000-kDa molecular mass cutoff. In parallel, ACE2-transfected cells were harvested and washed with PBS. For flow cytometry, $5.0 \times 10^5$ cells were resuspended in 1-mL incubation buffer [Dulbecco's PBS containing 0.02% EDTA (Sigma-Aldrich, St. Louis, Missouri, USA), 50 μg/mL DNase I (Worthington), and 5 mM $MgCl_2$] and incubated with 20 ug/mL of RBD-Fc for 30 minutes at room temperature as previously described [22]. Cells were then washed with buffer (5% FBS, 0.1% sodium azide in PBS) and incubated with human ACE2 Alexa Fluro 647-conjugated antibody (1 μg/$10^6$ cell, #FAB9332R, R&D Systems Minneapolis, Minnesota, USA), human IgG Fc PE-conjugated antibody (10μg/$10^6$ cell, #FAB110P, R&D Systems), or human IgG Fc APC-conjugated antibody (10μg/$10^6$ cell, #FAB110A, R&D Systems) for 30 minutes at room temperature. LSR II (BD Bioscience, Franklin lakes, New Jersey, USA) was used to collect the data, and Flowjo (Flowjo, Ashland, Oregon, USA) was used for analysis, conducted as previously described [22]. For immunoprecipitation, ACE2-transfected cells were lysed in a PBS buffer containing 1% CHASPO (Sigma-Aldrich) and incubated with Dynabeads Protein G (Thermo Fisher) and 2 μg of RBD-Fc concentrate. Dynabeads were washed with PBS (0.5% CHAPSO), and elutions were immunoblotted using antibodies against 1D4 (Abcam; ab5417) and Human IgG Fc (Abcam, Cambridge, UK; ab97225).

## Pseudovirus assay

SARS-CoV-2 spike pseudotyped MLV were generated based on a published protocol [105]. Briefly, HEK293T cells were transfected with 3 plasmids, which express MLV Gag and Pol, firefly luciferase reporter and SARS-CoV-2 S protein. Forty-eight hours after transfection, culture medium containing pseudotyped particles were centrifuged and then filtered through a sterile 0.45-μm pore-sized filter to remove cell debris. For transduction, HEK293T cells were first transfected with ACE2 orthologs using lipofectamine 3000. Twenty-four hours later, 200 μL of SARS-CoV-2 Spike pseudotyped MLV were added to each well and incubated for 2 days. The transduction efficiency was quantified by measuring the activity of luciferase using luciferase assay system (Promega, Madison, Wisconsin, USA) and GloMax 20/20 luminometer (Promega).

## ACE2 hydrolysis assay

HEK293T cells were transfected with ACE2 constructs using TransIT-X2 (Mirus). Seventy-two hours later, cells were washed in PBS and lysed in ACE2 reaction buffer pH 6.5 (1 M NaCl, 0.5% Triton X-100, 0.5 mM $ZnCl_2$, 75 mM Tris-HCl). Cell lysates were diluted in reaction buffer to 0.5-μg protein, incubated ± the ACE2-specific inhibitor 10μM DX600 (Cayman Chemical, Ann Arbor, Michigan, USA) for 20 minutes at room temperature, followed by addition of 100 μM Mca-YVADAPK(Dnp) (R&D Systems) and incubation at 37˚C. Fluorescence emission at 405 nm was measured at 10, 60, and 120 minutes using a microplate reader (BMG Labtech, Ortenberg, Germany) after excitation at 320 nm. Hydrolysis rates were quantified as fluorescence units per minute, using the slope of fluorescence development between 10 and 120 minutes, as previously described [47]. An ANOVA general linear model with fluorescence as the response variable, mutation as the factor, and assay time as the covariate was fit to fluorescence data generated in the ACE2 hydrolysis assay. Statistical differences in ACE2 hydrolysis rates were determined using a cross factor between the mutation factor and time covariate.

## Phylogenetic comparative methods

Human ACE and ACE2 coding sequences were used as blast queries to identify mammalian ACE2 orthologs. After removing low-quality sequences with gaps and ambiguous characters, this resulted in 107 species representing all major jawed vertebrate lineages were obtained from GenBank (S1 Table). This encompassed nearly the entire ACE2 coding sequence (residues 22 to 742, human ACE2 numbering; uniport ID Q9BYF1). These sequences were aligned using PRANK followed by manual adjustment [106]. This alignment was used to estimate a gene tree using PhyML 3.1 [107] (S5 Fig), with GTR selected using automatic model selection based on AIC values [108], and aLRT SH-like branch support. This ML tree was rooted using *ACE* and recapitulated all major phylogenetic relationships (S4 Fig) [109]. Ancestral sequences and posterior probability distributions were inferred using the best fitting models in the codeml package of PAML 4.9 (S8 Table) [58]. For estimation of dN/dS values using random sites models in the codeml package [58] and HyPhy [60,110], ACE sequences were pruned from the dataset, nonmammalian *ACE2* was also pruned, and additional mammalian *ACE2* sequences added to increase sampling. This alignment represented 89 species (S1 Table) and was used to infer a ML gene tree using IQ-Tree (S2 Fig), with the substitution model auto selected, followed by ultrafast bootstrap analysis and SH-aLRT branch tests [111]. PAML random sites models were used to investigate evidence of positive selection (S2 Table). To test for dN/dS differences among branches in the phylogeny, clade model D (CmD) [112] was used to analyze an *ACE2* alignment containing only those sites of interest in the ACE2 viral binding interface (S6 Table). M3 with 3 site classes was used as the null model for CmD. All random

sites and clade model PAML model pairs were statistically evaluated for significance by likelihood ratio tests (LRTs) with a $\chi 2$ distribution.

We conducted a phylogenetic comparative analysis on systolic blood pressure dataset by combining bat data with mammalian data compiled from a previous study [67,113]. We pruned this dataset to include only species with fossil calibrated divergence times [109] (S6 Fig). Body size values (grams) were obtained from the Ageing Genomic Resources AnAge database [114]. We conducted a phylogenetically independent correlation analysis (least squares linear regression; S7 Fig) by using the PDTREE program of the PDAP module of MESQUITE [115,116] to calculate phylogenetically independent contrasts of $\log_{10}$ body mass (grams) with systolic blood pressures (mm Hg) as the dependent variable. Before independent contrasts were calculated, branch lengths reflecting divergence times were ln transformed to meet the assumptions of independent contrast analysis [115,116]. To produce dN/dS estimates for branches in the untransformed phylogeny, we pruned the mammalian *ACE2* alignment to match the species represented in the blood pressure dataset and subjected the phylogeny and the alignment to analysis by aBSREL [117].

## Supporting information

**S1 Fig. Recombinant ACE2 hydrolysis activity in solubilized transfected HEK293T cells.** ACE2 hydrolysis activity was measured using a fluorogenic peptide substrate (Mca-YVA-DAPK(Dnp)-OH) incubated with lysates of HEK293T cells for 2 hours. Cells were either untransfected or were transfected with a human ACE2 construct. Incubation of lysate–peptide mixture with a ACE2-specific inhibitor (DX600) reduced fluorescence attributable to ACE2 hydrolysis activity. $N$ = 5 to 6 biological replicates. Standard error is shown. All data are available in S1 Data. ACE2, angiotensin converting enzyme 2.
(DOCX)

**S2 Fig. Conservation of angiotensin peptide sequences across mammalian species investigated in this study.** Renin produces angiotensin 1 by cleaving Angiotensinogen (*AGT* gene). Angiotensin 1 is subsequently cleaved by ACE, followed by ACE2. ACE2, angiotensin converting enzyme 2.
(DOCX)

**S3 Fig. Expression of ACE2–gfp orthologs in transfected HEK293T cells was assessed by flow cytometry (% of GFP-positive cells).** These expressed ACE2 proteins were used in hydrolysis assays. Human ACE2 served as an internal control in each separate assay. ACE2, angiotensin converting enzyme 2.
(DOCX)

**S4 Fig. Maximum likelihood phylogeny used in PAML analyses.** aLRT-SH like branch support values (IQ-Tree) are shown. Note that a basal trichotomy was artificially induced to accommodate input file requirements. All data are available in S1 Data.
(DOCX)

**S5 Fig. Targeted mutations to human ACE2 disrupt binding to the RBD of SARS-CoV-1 and SARS-CoV-2.** Western blots of immunopreciptations and cell lysates of HEK293T cells co-transfected with an Fc-tagged SARS-CoV-2 S protein RBD and 1D4-tagged (C9) human ACE2 construct. ACE2, angiotensin converting enzyme 2; RBD, receptor-binding domain; SARS-CoV-1, Severe Acute Respiratory Syndrome Coronavirus; SARS-CoV-2, Severe Acute Respiratory Syndrome Coronavirus 2.
(DOCX)

**S6 Fig. Maximum likelihood phylogeny used in ancestral reconstruction of Rodent ACE2.** aLRT-SH like branch support values (PhyML) are shown. All data are available in S1 Data. ACE2, angiotensin converting enzyme 2.
(DOCX)

**S7 Fig. Species phylogeny and least squares linear regression using phylogenetically independent contrasts of systolic blood pressure and body mass.** All data are available in S1 Data.
(DOCX)

**S1 Table. ACE2 accession numbers used in dN/dS estimates.** ACE2, angiotensin converting enzyme 2.
(DOCX)

**S2 Table. Analyses of selection on mammalian *ACE2* using PAML random sites models.** ACE2, angiotensin converting enzyme 2.
(DOCX)

**S3 Table. Results of BUSTED (HyPhy) analyses of mammalian *ACE2*.** This model accounts for synonymous rate variation (SRV). *Log* L values demonstrate that the unconstrained model performs better than the constrained, specifically due to the inclusion of a positive selection omega site category ($\omega_3$). ACE2, angiotensin converting enzyme 2.
(DOCX)

**S4 Table. Analyses of selection on mammalian *ACE2* without bat (Chioptera) sequences using PAML random sites models.** ACE2, angiotensin converting enzyme 2.
(DOCX)

**S5 Table. Positively selected sites in mammalian ACE2.** ACE2, angiotensin converting enzyme 2.
(DOCX)

**S6 Table. Results of CmD analyses of mammalian ACE2 under various partitions.** ACE2, angiotensin converting enzyme 2; CmD, clade model D.
(DOCX)

**S7 Table. ACE and ACE2 accession numbers used in ancestral reconstruction.** ACE, angiotensin converting enzyme; ACE2, angiotensin converting enzyme 2.
(DOCX)

**S8 Table. Results of random sites analyses of vertebrate *ACE2*, with the best fitting mode (M8) used for the ancestral reconstruction of mammalian ACE2.** ACE2, angiotensin converting enzyme 2.
(DOCX)

**S1 Data. Contains all individual data points used to derive the means and errors represented throughout this study.**
(XLSX)

## Acknowledgments

The authors thank the Johns Hopkins Genetic Resources Core Facility (RRID:SCR_018669) for DNA sequencing.

## Author Contributions

**Conceptualization:** Gianni M. Castiglione, Elia J. Duh.

**Data curation:** Gianni M. Castiglione.

**Formal analysis:** Gianni M. Castiglione, Lingli Zhou, Zhenhua Xu.

**Funding acquisition:** Elia J. Duh.

**Investigation:** Gianni M. Castiglione, Lingli Zhou, Zhenhua Xu, Zachary Neiman, Chien-Fu Hung, Elia J. Duh.

**Methodology:** Gianni M. Castiglione, Lingli Zhou, Zhenhua Xu, Chien-Fu Hung, Elia J. Duh.

**Project administration:** Elia J. Duh.

**Resources:** Elia J. Duh.

**Supervision:** Elia J. Duh.

**Validation:** Gianni M. Castiglione, Lingli Zhou, Zhenhua Xu.

**Visualization:** Gianni M. Castiglione, Lingli Zhou.

**Writing – original draft:** Gianni M. Castiglione.

**Writing – review & editing:** Gianni M. Castiglione, Elia J. Duh.

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
