## [Editor Report · Decision Letter 0]

5 Mar 2021

Dear Dr Duh, 

Thank you for submitting your manuscript entitled "Ancient epistasis gave rise to the modern-day SARS-CoV-2 pandemic" for consideration as a Research Article by PLOS Biology.

Your manuscript has now been evaluated by the PLOS Biology editorial staff, as well as by an academic editor with relevant expertise, and I'm writing to let you know that we would like to send your submission out for external peer review.

Please re-submit your manuscript within two working days, i.e. by Mar 09 2021 11:59PM.

Kind regards,

Roli Roberts

Senior Editor

PLOS Biology

---

## [Decision Letter · Decision Letter 1]

11 May 2021

Dear Dr Duh,

Thank you very much for submitting your manuscript "Ancient epistasis gave rise to the modern-day SARS-CoV-2 pandemic" as a Research Article for review by PLOS Biology. As with all papers reviewed by the journal, yours was assessed and discussed by the PLOS Biology editors, by an academic editor with relevant expertise, and in this case by three independent reviewers. We had recruited a fourth reviewer, but they are very overdue and are not responding to communications, so we reached a decision using the three reviews received. Based on the reviews, I regret that we will not be pursuing this manuscript for publication in the journal.

You'll see while all three reviewers are positive about the research question, reviewer #3 raises a number of very serious concerns about the support for your central claims. Reviewer #1's assessment is more positive, but overlaps somewhat, with concerns about the logic underlying some of your arguments. All three reviewers feel that the claims would need to be tempered (to varying degrees) before publication. Unfortunately we were strongly persuaded by the careful analysis from reviewer #3, and will therefore not be considering your manuscript further.

The reviews are attached, and we hope they may help you should you decide to revise the manuscript for submission elsewhere. I am sorry that we cannot be more positive on this occasion. 

I hope you appreciate the reasons for this decision and will consider PLOS Biology for other submissions in the future. Thank you for your support of PLOS and of Open Access publishing.

Sincerely,

Roli Roberts

Roland Roberts

Senior Editor

PLOS Biology

rroberts@plos.org

REVIEWERS' COMMENTS:

Reviewer #1:

In this paper, Castiglione et al. use computational evolutionary and functional experiments to understand how sequence variation in mammalian ACE2 impacts this proteins enzymatic function in addition to its latent capacity to enable SARS-related coronavirus entry. The manuscript is quite nice, I like the figure graphics, and there are some interesting analyses. I really liked the work on epistasis, and also appreciated the conclusion that because of epistasis, simple predictions of what ACE2s a virus may or may not bind are not easy to predict from knowledge within a single ACE2 alone. I do not completely follow some of the logical links between different experiments, and I detail some minor concerns below. Nonetheless, there are some very interesting components in the manuscript that I find compelling and of broad potential interest.

Major points:

1. Title: I don't find the language of "gave rise to" the SARS-CoV-2 pandemic as appropriately describing the findings here. This title suggests a much more proximal role in this ancient genetic variation and the origin of the pandemic. A better title might use language more like "enables susceptibility to SARS-CoV-2 binding"

2. Some inaccuracies or statements in describing the state of the field on SARS-CoV-2 evolution that could use correcting or more description:

a. Line 50: RaTG13 is not an "ancestor," but simply another tip on the phylogeny that has been sampled within the same sarbecovirus clade as SARS-CoV-2

b. Line 130-131: simply observing that pangolin ACE2 can bind SARS-CoV-2 does not "affirm" any intermediate host status for the direct SARS-CoV-2 line of transmission, which will end up being as much about ecology and contingency as much as anything else. (Though of course, binding/infectivity is a pre-requisitie, observing this binding/infectivity does not mean any individual species is an "intermediate" in the zoonotic sense). There is not yet widespread understanding or evidence that pangolins are direct intermediates in the spillover of SARS-CoV-2 itself, let alone being a result to be "reaffirmed" as in the language here

c. Line 125: there are multiple alleles of R. sinicus ACE2 (e.g. PMID 32699095) that differ in their SARS1 binding capacity - which sequence is being used here should be explicitly stated when introducing the sequence in the main text. And in general, making it clear you tested "a" bat/Rs ACE2 sequence across all language seems important to appropriately represent that there are many different bat ACE2s, including many different alleles even within this one host species

3. I have a couple important questions about the ACE2 hydrolysis activity assays:

a. Is the ACE2 peptide that is being hydrolyzed 100% conserved across the species being tested? If not, that would raise a major caveat about the assays as currently performed and their physiological relevance

b. Is the raw expression of ACE2 among variants within the cell lysates used for activity assays a potential confounder of the measured activities? If so, it may be necessary to quantify and normalize activity by relative expression levels in order to make conclusions about differences in activity of these orthologs.

4. Fig. 2F-G, lines 225-227: I do not interpret the difference between the effect of the K353H mutation as measured individually versus on top of the 79/82/83/84 mutations as exhibiting a "more muted effect" in the background of the additional mutations as stated in the manuscript. I think this may be a consequence of me not understanding the proper "scale" that mutations should combine on if additive/non-epistatic. For example, K353H alone has a 50% reduction in the relative RBD association metric - which could also be described as a 2-fold loss of binding relative to WT. When K353H is introduced on top of the 79/82/83/84 mutant background, it has only a 20-30% "raw" decrease in RBD binding (my understanding is this is what the authors are describing as "more muted"), but in fact, this 20-30% decrease in RBD association is >2-fold loss of binding relative to the 79/82/83/84 mutant in both the human and dog ACE2 backgrounds - which would argue the opposite of the authors conclusion, with K353H having a larger relative effect in this combinatorial background. I think in either case, given this assay is not a quantitative binding assay with e.g. thermodynamic measurements, we can't truly know how mutations would additively combine w.r.t. this metric, and so probably making any conclusiosn about potential magnitude epistasis is not fruitful. (This does not impact the sign epistasis in regards to the ACE2 activity assays, as sign epistasis is not subject to this 'uncertain scale of additivity' conern.)

5. Lines 163-166; 177-179; 182-184 : for the tests of positive selection, are you detecting positive selection within specific branches on the tree, or is it just saying "there is postivie selection somewhere on this tree"? Several publications have illustrated positive selection in bats, especially within the Rhinolophus bat reservoirs of sarbecoviruses, which is presumably due to selection on ACE2 sequence specifically to evade sarbecovirus binding and infection. (PMID 32699095, 22438550). In the current manuscript, it is unclear to me whether this positive selection within bats specifically due to viral pressures is giving rise to the positive selection signal, which is then being interpreted across all mammalian orders as evidence for selection related to intrinsic ACE2 function. This should be clarified, and probably the statements implying that this positive selection is specifically due to ACE2 physiological enzymatic activity might need to be tempered. (And these two papers are probably worth discussing specifically in the context of this work including the ACE2 residues they identify as positively selected in these host-virus arms' races.)

6. Line 280: it is unclear to me why having a lower native blood pressure would relax selection on ACE2 function. Regardless of the homeostatic 'set point', presumably the enzymatic activity of ACE2 is needed anyway to maintain that homeostatic set point. It seems like changes in global blood pressure are probably instead modulated by e.g. the upstream regulation of the peptide itself, or response pathways to the cleaved product, or some other pathway - not the actual catalytic activity of ACE2 itself. And it's further unclear why a correlation in between body mass and blood pressure establishes any relaxation in constraint - it's actually almost the opposite, in that it argues that there is some overarching 'reason' why smaller animals have lower blood pressure and therefore in fact it is an attuned process, not a relaxation of constraint. This analysis is interesting, but I think the authors might need to more carefully consider how to link it into their overall 'thesis'. The further linkage of all of this to effective population size in lines 369-371 extends this all way too far in my opinion, and the statements on lines 369-371 should just be left out.

7. The setup argued for starting in line 294 does not seem to be 'followed through' with the ASR that was actually performed. Simple measurement of the activity of the reconstructed sequence does not clearly illustrate whether these mutations were differentially tolerated in this ancestor without the loss of activity. It seems this analysis should involve not only reconstructing the ancestor, but also introducing the same mutations as illustrated in Figs. 2F/G into this ancestral sequence to identify whether the 'valley' is absent in this ancestral rodent sequence. I understand that's asking for substantial additional experiment, but it would really increase the interest added by this ASR component to the story - without it, the ASR component doesn't seem to add much additional insight. 

Minor points:

1. Line 51: "fusing" with isn't clear what that means, 

2. Not sure if the use of "pleiotropic" on line 110 is necessarily wrong, but it is sort of tripping me up. Maybe the directionality is reversed? The argument is not that this surface has evolved as a SARS-CoV-2 interaction interface and has pleiotropic consequences for ACE2 activity, but rather sort of the reverse - that evolution of ACE2 sequences due to differences in physiology alters the latent capacity for sarbecovirus receptor usage and susceptibility. Somehow describing this more specifically might enable the authors to avoid using a somewhat contentious word of 'pleiotropy' loosely

3. The disconcordance between Fig. 2F/G and Fig. S4 seems important for proper assessment of results. It suggests that the dynamic range of the RBD binding assay is lower than for actual viral entry. It might be worth simply including Fig. S4 within Fig. 2

Reviewer #2:

In this fascinating article, the authors undertake a series of investigations into the broad host tropism exhibited by SARS-CoV-2, beyond that predicted by studies focused exclusively on comparative sequence analyses of the ACE2 homology to the human ACE2 viral binding interface. In particular, the authors undertake the following major analyses:

1.They investigate ACE2 to SARS-CoV-1 (hereafter SC1) and SARS-CoV-2 (hereafter SC2) RBD binding across a suite of mammalian ACE2 orthologs transfected into 293T cells: human, mouse, bat, dog, pangolin. They show stronger association of SC2-RBD with both human and pangolin ACE2 relative to SC1 and show that the SC1 and SC2 proteins cannot bind mouse not bat ACE2

2.They follow this up in a pseudotype virus system, showing cell entry patterns that recapitulate those described in #1 across these same 5 transfected cell lines.

3.They then measured functional variation in ACE2 enzymatic activity across these same cell lines and found a wide range of hydrolysis activity. Dog, bat, and pangolin displayed low activity compared to human and mouse.

4.They next investigated dN/sS shifts in ACE2 mutational rates across these ACE2 orthologs in a suite of mammals and found evidence of ACE2 under positive selection. 

5.They next identified six residues unique to mouse ACE2 and adjacent to the viral RBD to explore the effects of these on virus binding. They took the same approach as in #4 to show that "several" of the sites were under either positive or purifying selection and that primates demonstrated decreased evolutionary rates relative to bats or rodents. 

6.They then used site-directed mutagenesis to substitute these six mouse sites in human and dog ACE2. They demonstrate that single mutations had limited effects on SC2 binding and huge deleterious effects on ACE2 hydrolysis activity but when introduced altogether both abolished AC2 binding and rescued the hydrolysis activity of ACE2 due to epistatic activity ('sign epistasis').

7.Now, the authors reasoned that species with higher blood pressure would have a greater need for ACE2 blood pressure regulation and therefore have greater constraints on ACE2 sequence space, so they attempted to correlate ACE2 evolutionary rate with blood pressure. They found that rodents and bats with higher ACE2 evolutionary rates also had lower blood pressure, possibly a mechanism for a less constrained ACE2 fitness landscape in these taxa

8.Finally, in order to explore the hypothesis that rodents may simply have evolved these six mutations by traversing a landscape of permissive mutations, the authors reconstructed ancestral rodent ACE2 and discovered it lacked the six residues of interest (suggesting these mutations evolved recently) but likely had high activity which could have compensated for deleterious mutations on the path to the modern receptor.

The paper is a tour-de-force of intriguing ideas and analyses. There are a few places where I have questions and where the discussion goes a bit too far but on the whole, it is in excellent shape:

1.For #4 above, why were the 107 chosen species selected (or the 89 included in the pruned analyses)? It seems likely that there may be extensive variation in dN/dS across bat species for instance. Some justification for the selected subset is needed

2.Following on above, the authors say that "several" of the six residues of interest for mouse ACE2 show evidence of positive or purifying selection across many taxa, with much higher rates in bats and rodents and constraints in humans. Can these findings be summarized in an accessible way, maybe in the supplement?

3.Can you make the y-axis on a log scale in Figure 3E? It's unclear whether there is any correlation at all or simply whether rodents and bats just show unusually high evolutionary rates for ACE2 compared with other taxa.

4.The authors talk a lot about bats and mice being "immune" to WT SC2. This is not known for bats. The authors would need to demonstrate lack in infectivity in a live animal model to show this, and in fact, they only show lack of virus entry into a transfected 293T cell line with a single bat species ACE2 ortholog. Since the bat chosen (R. sinicus) is not even the host for the closest known CoV to SC2 (R. affinis), this is a stretch. Additionally, Zhou et al 2020 shows some SC2 virus entry in HeLa cells expressing R. sinicus ACE2. Most of the work in this paper is focused on mouse immunity to SC2 and the relevant residues driving this interaction. I suggest the authors keep most of their speculations limited to mice and not try to extrapolate too far into bats.

5.Following on above, I would like to see a 'caveats and limitations' paragraph that mentions how we cannot really determine host range with a limited tool kit in this way. No mention is made of the fact that bats, for instance, might permit SC2 virus entry in some tissues (e.g. GIT tissues) and not others and might have other receptors permitting entry.

6.Additionally, the pangolin fusion hypotheses should be dropped. While pangolin CoV may effectively invade human cells, this paper provides no evidence that it is an intermediate host between bat CoV and WT SC2. It is largely believed that a closer genotype to SC2 than has yet been described is probably circulating in wild bats somewhere still. See MacLeean et al 2021, Boni et al 2020, Andersen et al 2020

7.Finally, the bit about genome engineering of minks or other domestic mammals should be dropped, as this is a big step, of questionable ethics, and moreover, this paper does not demonstrate that it would even work.

Reviewer #3:

This paper seeks to identify amino acid sequence states in the ACE2 protein that confer differences in susceptibility to SARS-Cov2 infection between species and to explain the evolution of these different states in terms of epistasis, natural selection, and pleiotropic effects on ACE2's endogenous enzyme activity.

Understanding the sequence-function relationships underlying ACE2 interactions with SARS-Cov2 is a worthy goal, as is understanding the evolutionary causes of differences in these interactions among species. The subject matter of the paper is therefore of significant potential interest. However, many of the claims are not supported by the experiments and analyses presented. I'm sorry to say that the claims that have sufficient support after a careful reading are of rather narrow scientific impact and seem best suited for a specialist audience. 

1. The authors' approach is to transfer 6 amino acid states that exist in mouse ACE2 into human and dog and measure their effects on molecular function. They choose these species and states because: 1) these residues are at sites on the surface of ACE2 that binds the SARS-Cov2 spike protein, 2) human and dog ACE2 have higher relative affinity for SARS-Cov2 than mouse ACE2 does, and 3) mouse is less susceptible to SARS-Cov2 infection than human and dog. With several nice experiments in Figs 1A-E, the authors provide evidence that reinforces premises 2 and 3: mouse ACE2 binds the SARS-Cov2 less efficiently than human and dog ACE2, and cultured cells transfected with mouse ACE2 are less susceptible to pseudovirus infection. 

2. But the paper does not show that the six residues are sufficient causes for the difference in affinity and infectivity between the species' ACE2 proteins. The major experiments transfer the 6 mouse states into human and dog ACE2 proteins. However, these "chimeric" ACE2 proteins do not confer cellular resistance to infection nearly as well as the mouse ACE2 protein itself does. Thus, the 6 mouse amino acids can reduce to some extent but not nearly recapitulate the fully resistant cellular phenotype conferred by mouse ACE2 (compare Fig 1E to Fig S4). Further, the reciprocal experiment, where human or dog amino acids at these sites are introduced into the mouse ACE2 was not performed; we therefore do not know the extent to which the 6 states account for the resistance exhibited by the mouse ACE2. Moreover, the historical substitutions from ancestral states to any derived state is never assayed in any of the species' proteins, as would be required to support the contention that these substitutions played a causal role in the evolution of increased or reduced affinity/susceptibility. The evidence therefore establishes that a small number of residues in mouse ACE2 can partially reduce affinity and infectivity when introduced into human or dog. But it does not support a causally sufficient role for the mouse states in resistance by the mouse ACE2. This observation is interesting in terms of ACE2 sequence-function relationships, but it does not have clearly interpretable implications for genetic/biochemical causality that underlies differences between species or the evolution of those differences.

3. A central claim of the paper is that the 6 states interact epistatically in producing the reduced affinity of ACE2 for SARS-Cov2 and the reduced enzyme activity. These claimed epistatic interactions are then said to explain why susceptible species have not evolved genotypes resistant to SARS-Cov2. The evidence for epistasis with respect to affinity is not convincing, because detecting epistasis requires a significant deviation from a well-founded expectation for a quantitative phenotype that would be observed in the absence of epistasis. For example, in the absence of epistasis, one would expect the effect of a combination of mutations on the free energy of binding to be the sum of the energetic effects of each mutation introduced singly; the effect on Kd is expected to be multiplicative. But no such expectation or test is provided here to show that the effects of combinations are different from the effects that would arise if there were no epistasis. For binding, the assay is a complex one that does not directly quantify affinity, occupancy of the bound state, etc. 

reasons. 

 The paper claims an epistatic interaction for binding between mutation at site 353 and those at the other sites, but there is no apparent epistasis at all on this case: site 353 and the set of the other mutations each reduce affinity, and they reduce affinity to a greater extent when combined, precisely as expected with no epistasis. The data do show that 5 of the 6 mouse states produce no clear effect on binding when introduced singly, and they do reduce binding when combined with each other or with the sixth state which affects affinity on its own. This does not necessarily indicate epistasis. Suppose the assay has an intrinsically nonlinear dose-response relationship (such as a hyperbolic or sigmoidal relationship, as is expected in any saturable assay); in such a case, single mutations that each have a moderate effect on affinity may produce no detectable reduction in the signal of binding, if introduced singly into a high-affinity protein, but when introduced into a protein whose affinity has already been weakened by other mutations, that effect will become apparent. Further, there is no evidence for epistasis in the infectivity assay shown in Fig S4, where progressively including more mutations progressively reduces infectivity. These observations do not rule out the possibility of some kind of relatively subtle epistasis with respect to the magnitude of mutations' effects, but they do not establish it. The paper therefore provides no persuasive evidence of epistasis for the phenotype of ACE2 affinity for SARS-Cov2 or susceptibility. 

 In the absence of quantitative knowledge concerning the expected phenotype when nonepistatic mutations are combined, one could still provide some evidence of epistasis if the sign of the effect of a mutation differs when introduced into different genetic backgrounds; the only way sign epistasis can arise without epistasis is for the underlying relationship between the measured phenotype and underlying biochemical effects to be nonmonotonic, and that is unlikely in the case of apparent binding in these assays. The authors do observe sign epistasis for the catalytic phenotype, because the mouse state at site 353 reduces activity on its own when introduced into human ACE2 but increases it when introduced into the context of four other mouse states. Thus, the authors should make no claim for epistasis with respect to binding or infectivity, but they can make a limited claim for epistasis of this mutational combination with respect to catalysis.

4. The paper's central evolutionary narrative is that the lack of evolved resistance in humans and dog is attributable to a claimed pleiotropic cost incurred by reducing ACE2's affinity for SARS-Cov2. Mice are claimed to be free of this pleiotropic constraint, because the function of ACE2 in their cardiovascular system is different from that in humans and many other mammals. The data do not coherently support this premise, for several reasons. 

 a. Fig. 1 shows that the mouse ACE2 actually has higher peptidase activity than human and dog, not lower, as would be required for the peptidase-versus-affinity tradeoff to explain the evolution of viral resistance in mouse but not in humans. 

reasons. 

 b. The authors observe reduced ACE2 enzyme activity when the 6 mouse states are introduced into the human ACE2, but no such reduction is observed in dog. This means that there is no intrinsic association between the two phenotypes, as is required to claim that humans and dogs have not evolved resistance to SARS-Cov2 because of antagonistic pleiotropy related to ACE2 activity. 

 c. The observation that the 6 mouse states decrease SARS-Cov2 affinity and reduce peptidase activity in human ACE2 would at best imply only that humans may be unlikely to evolve reduced affinity by acquiring these particular six mouse states. This does not establish that they could not do so via other mutations that may not affect peptidase activity. 

reasons. 

 d. The fact that the six mouse states do not have the deleterious effects on dog ACE2 indicates that the states at other sites in the protein can prevent the deleterious effect of the six mouse states, indicating that human ACE2 might be able to acquire the 6 mouse states if it also acquired other residues that have a similar modifying or buffering effect. Thus, the authors' data establish only that human ACE2 could not reduce its SARS-Cov2 affinity without incurring pleiotropic effects on affinity by acquiring only the 6 mouse sites. A general statement about acquisition of resistance per se therefore cannot be justified.

5. The authors claim that evolution of the ACE2 protein and several of the six states in particular has been driven by positive selection. They base this claim on the results of two kinds of model-based likelihood ratio test, the branch-sites test and the sites test. However, both of these tests have been shown in the literature to be unreliable, with very high propensities to return false positive conclusions under realistic conditions. These methods therefore do not provide reliable evidence for the claims about selection. It is true that these methods have been widely used in the past as evidence for positive selection; given the recent findings, however, they should no longer be used. See Witosky et al, Synonymous site-to-site substitution rate variation dramatically inflates false positive rates of selection analyses: ignore at your own peril, MBE 2020; Venkat et al, Multinucleotide mutations cause false inferences of lineage-specific positive selection, Nature Evol Evol 2018.

6. The authors claim that the 6 mouse states could have been selectively accessible in rodents because rodents have lower systolic blood pressure than other mammals, which could result in lower selective pressure to maintain ACE2 function, thereby reducing the deleterious costs of the 6 states. The authors provide as evidence of this relaxed constraint hypothesis a higher ratio of nonsynonymous to synonymous rates of evolution at these sites in rodents and bats compared to other mammals. In addition to concerns about these tests as discussed above, the analysis appears to have been performed only on the subset of sites that differ in amino acid state between mouse and other mammals, which are tautologically expected to have higher rates of nonsynonymous substitution in rodents. 

 A second problem is that the authors also state that unlike other mammals, mice do not exhibit a vasodilatory response to Ang-(1-7), the product of ACE2 hydrolysis. This observation seems to contradict the low blood pressure hypothesis for putative relaxed selective constraint: if ACE2 does not mediate vasodilation, then it is unclear why low blood pressure should produce any relaxed constraint at all.

7. Based on the observation that the 6 mouse states that reduce the peptidase activity of human ACE2 are not located at the protein's catalytic active site, the authors state that this effect must be mediated by indirect structural effects. However, the most plausible mechanism by which mutations would have this effect would be by impairing substrate binding, which takes place at the portion of the ACE2 surface where SARS-Cov2 binding occurs. This would not reflect a surprising mechanism, and it would not be indirect, as it would involve mutations at the protein's surface directly compromising interactions with the substrate at that surface. 

8. The authors state that it is surprising that the 6 mouse residues reduce SARS-Cov2 affinity when introduced into either human or dog, because human and dog have different states at most of these sites. They say that this indicates that homology-based reasoning is a poor predictor of proteins' affinity. But the observation that the human and dog states are different is not at all surprising - all it means is that there are multiple amino acid states per residue that are compatible will affinity higher than that conferred by the mouse states. It is very common in multiple sequence alignments to observe some sequence variability at functionally important sites in a protein, such as exchanges between hydrophobic states in a protein's core, or between polar states on a protein's surface, or between donor states (or acceptors) in hydrogen-bonding residues. No selective or epistatic explanation is required to account for this variation - some sequence degeneracy of the functional property is all that is required. The authors' results do show that a search for strict conservation of a single state between proteins with similar affinity is not a reliable guide to identifying sequence sites that contribute to that phenotype, but it would be very naïve to think that it would be. A further consideration related to sequence variability at these sites is that affinity for SARS-Cov2 could not possibly have been a source of long-term constraints that affect sequence variation among species, because the virus did not emerge until 2019. There is no reason that we should expect sites that contribute to affinity to be strictly conserved over evolutionary time.

9. The authors refer to a "functional convergence" between dog and human in their shared susceptibility to SARS-Cov2. But the paper suggests that susceptibility and high ACE2 affinity is ancestral, with a reduction in these phenotypes in the lineage leading to mouse. Susceptibility is therefore not convergent but a retained ancestral state.

I am sorry to say that when these issues are all considered, many of the paper's claims turn out not to be sufficiently supported by the evidence. The paper does establish that six states in mouse can contribute to reducing both affinity and peptidase activity when introduced into human ACE2; further, these states reduce affinity but not activity when introduced into dog ACE2. That is interesting from a sequence-function perspective and should be of interest to scientists whose studies are focused in detail on ACE2 binding and catalysis. But the current paper does not establish whether those states are sufficient to confer resistance in mouse, nor they establish why those states evolved in any of the taxa of interest. Further, there is virtually no evidence for epistasis or an effect on evolutionary processes. The idea of pleiotropic constraints contributing to susceptibility in many lineages is plausible, but the evidence presented does not support it. I do appreciate this paper's effort to connect genetic experiments in ACE2 to evolutionary processes, but the paper's claims on this subject are not justified by the evidence. In my opinion, then, the work reported here should be reported to specialists in the field, but in doing so the claims should be dramatically narrowed.

---

## [Editor Report · Decision Letter 2]

28 Sep 2021

Dear Dr Duh,

Thank you for your Appeal regarding our previous decision to reject your manuscript "Ancient epistasis gave rise to the modern-day SARS-CoV-2 pandemic" after peer-review. We appreciated the points that you raised, and your proposed revisions based on novel experimental data, and after discussion among the team and with the Academic Editor, we have decided to give you an opportunity to submit a much-revised version that takes into account the reviewers' comments.

However, we warn you that we cannot make any further decision until we have seen the revised manuscript and your response to the reviewers' comments. Your responses will need to satisfy the current reviewers, and it is possible that we may need to solicit further expert input in order to reach a firm decision.

We expect to receive your revised manuscript within 3 months. 

**IMPORTANT - SUBMITTING YOUR REVISION**

*Re-submission Checklist*

*Published Peer Review*

*PLOS Data Policy*

*Blot and Gel Data Policy*

Sincerely,

Roli Roberts

Roland Roberts

Senior Editor

PLOS Biology

rroberts@plos.org

REVIEWERS' COMMENTS:

Reviewer #1:

In this paper, Castiglione et al. use computational evolutionary and functional experiments to understand how sequence variation in mammalian ACE2 impacts this proteins enzymatic function in addition to its latent capacity to enable SARS-related coronavirus entry. The manuscript is quite nice, I like the figure graphics, and there are some interesting analyses. I really liked the work on epistasis, and also appreciated the conclusion that because of epistasis, simple predictions of what ACE2s a virus may or may not bind are not easy to predict from knowledge within a single ACE2 alone. I do not completely follow some of the logical links between different experiments, and I detail some minor concerns below. Nonetheless, there are some very interesting components in the manuscript that I find compelling and of broad potential interest.

Major points:

1. Title: I don't find the language of "gave rise to" the SARS-CoV-2 pandemic as appropriately describing the findings here. This title suggests a much more proximal role in this ancient genetic variation and the origin of the pandemic. A better title might use language more like "enables susceptibility to SARS-CoV-2 binding"

2. Some inaccuracies or statements in describing the state of the field on SARS-CoV-2 evolution that could use correcting or more description:

a. Line 50: RaTG13 is not an "ancestor," but simply another tip on the phylogeny that has been sampled within the same sarbecovirus clade as SARS-CoV-2

b. Line 130-131: simply observing that pangolin ACE2 can bind SARS-CoV-2 does not "affirm" any intermediate host status for the direct SARS-CoV-2 line of transmission, which will end up being as much about ecology and contingency as much as anything else. (Though of course, binding/infectivity is a pre-requisitie, observing this binding/infectivity does not mean any individual species is an "intermediate" in the zoonotic sense). There is not yet widespread understanding or evidence that pangolins are direct intermediates in the spillover of SARS-CoV-2 itself, let alone being a result to be "reaffirmed" as in the language here

c. Line 125: there are multiple alleles of R. sinicus ACE2 (e.g. PMID 32699095) that differ in their SARS1 binding capacity - which sequence is being used here should be explicitly stated when introducing the sequence in the main text. And in general, making it clear you tested "a" bat/Rs ACE2 sequence across all language seems important to appropriately represent that there are many different bat ACE2s, including many different alleles even within this one host species

3. I have a couple important questions about the ACE2 hydrolysis activity assays:

a. Is the ACE2 peptide that is being hydrolyzed 100% conserved across the species being tested? If not, that would raise a major caveat about the assays as currently performed and their physiological relevance

b. Is the raw expression of ACE2 among variants within the cell lysates used for activity assays a potential confounder of the measured activities? If so, it may be necessary to quantify and normalize activity by relative expression levels in order to make conclusions about differences in activity of these orthologs.

4. Fig. 2F-G, lines 225-227: I do not interpret the difference between the effect of the K353H mutation as measured individually versus on top of the 79/82/83/84 mutations as exhibiting a "more muted effect" in the background of the additional mutations as stated in the manuscript. I think this may be a consequence of me not understanding the proper "scale" that mutations should combine on if additive/non-epistatic. For example, K353H alone has a 50% reduction in the relative RBD association metric - which could also be described as a 2-fold loss of binding relative to WT. When K353H is introduced on top of the 79/82/83/84 mutant background, it has only a 20-30% "raw" decrease in RBD binding (my understanding is this is what the authors are describing as "more muted"), but in fact, this 20-30% decrease in RBD association is >2-fold loss of binding relative to the 79/82/83/84 mutant in both the human and dog ACE2 backgrounds - which would argue the opposite of the authors conclusion, with K353H having a larger relative effect in this combinatorial background. I think in either case, given this assay is not a quantitative binding assay with e.g. thermodynamic measurements, we can't truly know how mutations would additively combine w.r.t. this metric, and so probably making any conclusiosn about potential magnitude epistasis is not fruitful. (This does not impact the sign epistasis in regards to the ACE2 activity assays, as sign epistasis is not subject to this 'uncertain scale of additivity' conern.)

5. Lines 163-166; 177-179; 182-184 : for the tests of positive selection, are you detecting positive selection within specific branches on the tree, or is it just saying "there is postivie selection somewhere on this tree"? Several publications have illustrated positive selection in bats, especially within the Rhinolophus bat reservoirs of sarbecoviruses, which is presumably due to selection on ACE2 sequence specifically to evade sarbecovirus binding and infection. (PMID 32699095, 22438550). In the current manuscript, it is unclear to me whether this positive selection within bats specifically due to viral pressures is giving rise to the positive selection signal, which is then being interpreted across all mammalian orders as evidence for selection related to intrinsic ACE2 function. This should be clarified, and probably the statements implying that this positive selection is specifically due to ACE2 physiological enzymatic activity might need to be tempered. (And these two papers are probably worth discussing specifically in the context of this work including the ACE2 residues they identify as positively selected in these host-virus arms' races.)

6. Line 280: it is unclear to me why having a lower native blood pressure would relax selection on ACE2 function. Regardless of the homeostatic 'set point', presumably the enzymatic activity of ACE2 is needed anyway to maintain that homeostatic set point. It seems like changes in global blood pressure are probably instead modulated by e.g. the upstream regulation of the peptide itself, or response pathways to the cleaved product, or some other pathway - not the actual catalytic activity of ACE2 itself. And it's further unclear why a correlation in between body mass and blood pressure establishes any relaxation in constraint - it's actually almost the opposite, in that it argues that there is some overarching 'reason' why smaller animals have lower blood pressure and therefore in fact it is an attuned process, not a relaxation of constraint. This analysis is interesting, but I think the authors might need to more carefully consider how to link it into their overall 'thesis'. The further linkage of all of this to effective population size in lines 369-371 extends this all way too far in my opinion, and the statements on lines 369-371 should just be left out.

7. The setup argued for starting in line 294 does not seem to be 'followed through' with the ASR that was actually performed. Simple measurement of the activity of the reconstructed sequence does not clearly illustrate whether these mutations were differentially tolerated in this ancestor without the loss of activity. It seems this analysis should involve not only reconstructing the ancestor, but also introducing the same mutations as illustrated in Figs. 2F/G into this ancestral sequence to identify whether the 'valley' is absent in this ancestral rodent sequence. I understand that's asking for substantial additional experiment, but it would really increase the interest added by this ASR component to the story - without it, the ASR component doesn't seem to add much additional insight.

Minor points:

1. Line 51: "fusing" with isn't clear what that means,

2. Not sure if the use of "pleiotropic" on line 110 is necessarily wrong, but it is sort of tripping me up. Maybe the directionality is reversed? The argument is not that this surface has evolved as a SARS-CoV-2 interaction interface and has pleiotropic consequences for ACE2 activity, but rather sort of the reverse - that evolution of ACE2 sequences due to differences in physiology alters the latent capacity for sarbecovirus receptor usage and susceptibility. Somehow describing this more specifically might enable the authors to avoid using a somewhat contentious word of 'pleiotropy' loosely

3. The disconcordance between Fig. 2F/G and Fig. S4 seems important for proper assessment of results. It suggests that the dynamic range of the RBD binding assay is lower than for actual viral entry. It might be worth simply including Fig. S4 within Fig. 2

Reviewer #2:

In this fascinating article, the authors undertake a series of investigations into the broad host tropism exhibited by SARS-CoV-2, beyond that predicted by studies focused exclusively on comparative sequence analyses of the ACE2 homology to the human ACE2 viral binding interface. In particular, the authors undertake the following major analyses:

1.They investigate ACE2 to SARS-CoV-1 (hereafter SC1) and SARS-CoV-2 (hereafter SC2) RBD binding across a suite of mammalian ACE2 orthologs transfected into 293T cells: human, mouse, bat, dog, pangolin. They show stronger association of SC2-RBD with both human and pangolin ACE2 relative to SC1 and show that the SC1 and SC2 proteins cannot bind mouse not bat ACE2

2.They follow this up in a pseudotype virus system, showing cell entry patterns that recapitulate those described in #1 across these same 5 transfected cell lines.

3.They then measured functional variation in ACE2 enzymatic activity across these same cell lines and found a wide range of hydrolysis activity. Dog, bat, and pangolin displayed low activity compared to human and mouse.

4.They next investigated dN/sS shifts in ACE2 mutational rates across these ACE2 orthologs in a suite of mammals and found evidence of ACE2 under positive selection.

5.They next identified six residues unique to mouse ACE2 and adjacent to the viral RBD to explore the effects of these on virus binding. They took the same approach as in #4 to show that "several" of the sites were under either positive or purifying selection and that primates demonstrated decreased evolutionary rates relative to bats or rodents.

6.They then used site-directed mutagenesis to substitute these six mouse sites in human and dog ACE2. They demonstrate that single mutations had limited effects on SC2 binding and huge deleterious effects on ACE2 hydrolysis activity but when introduced altogether both abolished AC2 binding and rescued the hydrolysis activity of ACE2 due to epistatic activity ('sign epistasis').

7.Now, the authors reasoned that species with higher blood pressure would have a greater need for ACE2 blood pressure regulation and therefore have greater constraints on ACE2 sequence space, so they attempted to correlate ACE2 evolutionary rate with blood pressure. They found that rodents and bats with higher ACE2 evolutionary rates also had lower blood pressure, possibly a mechanism for a less constrained ACE2 fitness landscape in these taxa

8.Finally, in order to explore the hypothesis that rodents may simply have evolved these six mutations by traversing a landscape of permissive mutations, the authors reconstructed ancestral rodent ACE2 and discovered it lacked the six residues of interest (suggesting these mutations evolved recently) but likely had high activity which could have compensated for deleterious mutations on the path to the modern receptor.

The paper is a tour-de-force of intriguing ideas and analyses. There are a few places where I have questions and where the discussion goes a bit too far but on the whole, it is in excellent shape:

1.For #4 above, why were the 107 chosen species selected (or the 89 included in the pruned analyses)? It seems likely that there may be extensive variation in dN/dS across bat species for instance. Some justification for the selected subset is needed

2.Following on above, the authors say that "several" of the six residues of interest for mouse ACE2 show evidence of positive or purifying selection across many taxa, with much higher rates in bats and rodents and constraints in humans. Can these findings be summarized in an accessible way, maybe in the supplement?

3.Can you make the y-axis on a log scale in Figure 3E? It's unclear whether there is any correlation at all or simply whether rodents and bats just show unusually high evolutionary rates for ACE2 compared with other taxa.

4.The authors talk a lot about bats and mice being "immune" to WT SC2. This is not known for bats. The authors would need to demonstrate lack in infectivity in a live animal model to show this, and in fact, they only show lack of virus entry into a transfected 293T cell line with a single bat species ACE2 ortholog. Since the bat chosen (R. sinicus) is not even the host for the closest known CoV to SC2 (R. affinis), this is a stretch. Additionally, Zhou et al 2020 shows some SC2 virus entry in HeLa cells expressing R. sinicus ACE2. Most of the work in this paper is focused on mouse immunity to SC2 and the relevant residues driving this interaction. I suggest the authors keep most of their speculations limited to mice and not try to extrapolate too far into bats.

5.Following on above, I would like to see a 'caveats and limitations' paragraph that mentions how we cannot really determine host range with a limited tool kit in this way. No mention is made of the fact that bats, for instance, might permit SC2 virus entry in some tissues (e.g. GIT tissues) and not others and might have other receptors permitting entry.

6.Additionally, the pangolin fusion hypotheses should be dropped. While pangolin CoV may effectively invade human cells, this paper provides no evidence that it is an intermediate host between bat CoV and WT SC2. It is largely believed that a closer genotype to SC2 than has yet been described is probably circulating in wild bats somewhere still. See MacLeean et al 2021, Boni et al 2020, Andersen et al 2020

7.Finally, the bit about genome engineering of minks or other domestic mammals should be dropped, as this is a big step, of questionable ethics, and moreover, this paper does not demonstrate that it would even work.

Reviewer #3:

This paper seeks to identify amino acid sequence states in the ACE2 protein that confer differences in susceptibility to SARS-Cov2 infection between species and to explain the evolution of these different states in terms of epistasis, natural selection, and pleiotropic effects on ACE2's endogenous enzyme activity.

Understanding the sequence-function relationships underlying ACE2 interactions with SARS-Cov2 is a worthy goal, as is understanding the evolutionary causes of differences in these interactions among species. The subject matter of the paper is therefore of significant potential interest. However, many of the claims are not supported by the experiments and analyses presented. I'm sorry to say that the claims that have sufficient support after a careful reading are of rather narrow scientific impact and seem best suited for a specialist audience.

1. The authors' approach is to transfer 6 amino acid states that exist in mouse ACE2 into human and dog and measure their effects on molecular function. They choose these species and states because: 1) these residues are at sites on the surface of ACE2 that binds the SARS-Cov2 spike protein, 2) human and dog ACE2 have higher relative affinity for SARS-Cov2 than mouse ACE2 does, and 3) mouse is less susceptible to SARS-Cov2 infection than human and dog. With several nice experiments in Figs 1A-E, the authors provide evidence that reinforces premises 2 and 3: mouse ACE2 binds the SARS-Cov2 less efficiently than human and dog ACE2, and cultured cells transfected with mouse ACE2 are less susceptible to pseudovirus infection.

2. But the paper does not show that the six residues are sufficient causes for the difference in affinity and infectivity between the species' ACE2 proteins. The major experiments transfer the 6 mouse states into human and dog ACE2 proteins. However, these "chimeric" ACE2 proteins do not confer cellular resistance to infection nearly as well as the mouse ACE2 protein itself does. Thus, the 6 mouse amino acids can reduce to some extent but not nearly recapitulate the fully resistant cellular phenotype conferred by mouse ACE2 (compare Fig 1E to Fig S4). Further, the reciprocal experiment, where human or dog amino acids at these sites are introduced into the mouse ACE2 was not performed; we therefore do not know the extent to which the 6 states account for the resistance exhibited by the mouse ACE2. Moreover, the historical substitutions from ancestral states to any derived state is never assayed in any of the species' proteins, as would be required to support the contention that these substitutions played a causal role in the evolution of increased or reduced affinity/susceptibility. The evidence therefore establishes that a small number of residues in mouse ACE2 can partially reduce affinity and infectivity when introduced into human or dog. But it does not support a causally sufficient role for the mouse states in resistance by the mouse ACE2. This observation is interesting in terms of ACE2 sequence-function relationships, but it does not have clearly interpretable implications for genetic/biochemical causality that underlies differences between species or the evolution of those differences.

3. A central claim of the paper is that the 6 states interact epistatically in producing the reduced affinity of ACE2 for SARS-Cov2 and the reduced enzyme activity. These claimed epistatic interactions are then said to explain why susceptible species have not evolved genotypes resistant to SARS-Cov2. The evidence for epistasis with respect to affinity is not convincing, because detecting epistasis requires a significant deviation from a well-founded expectation for a quantitative phenotype that would be observed in the absence of epistasis. For example, in the absence of epistasis, one would expect the effect of a combination of mutations on the free energy of binding to be the sum of the energetic effects of each mutation introduced singly; the effect on Kd is expected to be multiplicative. But no such expectation or test is provided here to show that the effects of combinations are different from the effects that would arise if there were no epistasis. For binding, the assay is a complex one that does not directly quantify affinity, occupancy of the bound state, etc.

reasons.

The paper claims an epistatic interaction for binding between mutation at site 353 and those at the other sites, but there is no apparent epistasis at all on this case: site 353 and the set of the other mutations each reduce affinity, and they reduce affinity to a greater extent when combined, precisely as expected with no epistasis. The data do show that 5 of the 6 mouse states produce no clear effect on binding when introduced singly, and they do reduce binding when combined with each other or with the sixth state which affects affinity on its own. This does not necessarily indicate epistasis. Suppose the assay has an intrinsically nonlinear dose-response relationship (such as a hyperbolic or sigmoidal relationship, as is expected in any saturable assay); in such a case, single mutations that each have a moderate effect on affinity may produce no detectable reduction in the signal of binding, if introduced singly into a high-affinity protein, but when introduced into a protein whose affinity has already been weakened by other mutations, that effect will become apparent. Further, there is no evidence for epistasis in the infectivity assay shown in Fig S4, where progressively including more mutations progressively reduces infectivity. These observations do not rule out the possibility of some kind of relatively subtle epistasis with respect to the magnitude of mutations' effects, but they do not establish it. The paper therefore provides no persuasive evidence of epistasis for the phenotype of ACE2 affinity for SARS-Cov2 or susceptibility.

In the absence of quantitative knowledge concerning the expected phenotype when nonepistatic mutations are combined, one could still provide some evidence of epistasis if the sign of the effect of a mutation differs when introduced into different genetic backgrounds; the only way sign epistasis can arise without epistasis is for the underlying relationship between the measured phenotype and underlying biochemical effects to be nonmonotonic, and that is unlikely in the case of apparent binding in these assays. The authors do observe sign epistasis for the catalytic phenotype, because the mouse state at site 353 reduces activity on its own when introduced into human ACE2 but increases it when introduced into the context of four other mouse states. Thus, the authors should make no claim for epistasis with respect to binding or infectivity, but they can make a limited claim for epistasis of this mutational combination with respect to catalysis.

4. The paper's central evolutionary narrative is that the lack of evolved resistance in humans and dog is attributable to a claimed pleiotropic cost incurred by reducing ACE2's affinity for SARS-Cov2. Mice are claimed to be free of this pleiotropic constraint, because the function of ACE2 in their cardiovascular system is different from that in humans and many other mammals. The data do not coherently support this premise, for several reasons.

a. Fig. 1 shows that the mouse ACE2 actually has higher peptidase activity than human and dog, not lower, as would be required for the peptidase-versus-affinity tradeoff to explain the evolution of viral resistance in mouse but not in humans.

reasons.

b. The authors observe reduced ACE2 enzyme activity when the 6 mouse states are introduced into the human ACE2, but no such reduction is observed in dog. This means that there is no intrinsic association between the two phenotypes, as is required to claim that humans and dogs have not evolved resistance to SARS-Cov2 because of antagonistic pleiotropy related to ACE2 activity.

c. The observation that the 6 mouse states decrease SARS-Cov2 affinity and reduce peptidase activity in human ACE2 would at best imply only that humans may be unlikely to evolve reduced affinity by acquiring these particular six mouse states. This does not establish that they could not do so via other mutations that may not affect peptidase activity.

reasons.

d. The fact that the six mouse states do not have the deleterious effects on dog ACE2 indicates that the states at other sites in the protein can prevent the deleterious effect of the six mouse states, indicating that human ACE2 might be able to acquire the 6 mouse states if it also acquired other residues that have a similar modifying or buffering effect. Thus, the authors' data establish only that human ACE2 could not reduce its SARS-Cov2 affinity without incurring pleiotropic effects on affinity by acquiring only the 6 mouse sites. A general statement about acquisition of resistance per se therefore cannot be justified.

5. The authors claim that evolution of the ACE2 protein and several of the six states in particular has been driven by positive selection. They base this claim on the results of two kinds of model-based likelihood ratio test, the branch-sites test and the sites test. However, both of these tests have been shown in the literature to be unreliable, with very high propensities to return false positive conclusions under realistic conditions. These methods therefore do not provide reliable evidence for the claims about selection. It is true that these methods have been widely used in the past as evidence for positive selection; given the recent findings, however, they should no longer be used. See Witosky et al, Synonymous site-to-site substitution rate variation dramatically inflates false positive rates of selection analyses: ignore at your own peril, MBE 2020; Venkat et al, Multinucleotide mutations cause false inferences of lineage-specific positive selection, Nature Evol Evol 2018.

6. The authors claim that the 6 mouse states could have been selectively accessible in rodents because rodents have lower systolic blood pressure than other mammals, which could result in lower selective pressure to maintain ACE2 function, thereby reducing the deleterious costs of the 6 states. The authors provide as evidence of this relaxed constraint hypothesis a higher ratio of nonsynonymous to synonymous rates of evolution at these sites in rodents and bats compared to other mammals. In addition to concerns about these tests as discussed above, the analysis appears to have been performed only on the subset of sites that differ in amino acid state between mouse and other mammals, which are tautologically expected to have higher rates of nonsynonymous substitution in rodents.

A second problem is that the authors also state that unlike other mammals, mice do not exhibit a vasodilatory response to Ang-(1-7), the product of ACE2 hydrolysis. This observation seems to contradict the low blood pressure hypothesis for putative relaxed selective constraint: if ACE2 does not mediate vasodilation, then it is unclear why low blood pressure should produce any relaxed constraint at all.

7. Based on the observation that the 6 mouse states that reduce the peptidase activity of human ACE2 are not located at the protein's catalytic active site, the authors state that this effect must be mediated by indirect structural effects. However, the most plausible mechanism by which mutations would have this effect would be by impairing substrate binding, which takes place at the portion of the ACE2 surface where SARS-Cov2 binding occurs. This would not reflect a surprising mechanism, and it would not be indirect, as it would involve mutations at the protein's surface directly compromising interactions with the substrate at that surface.

8. The authors state that it is surprising that the 6 mouse residues reduce SARS-Cov2 affinity when introduced into either human or dog, because human and dog have different states at most of these sites. They say that this indicates that homology-based reasoning is a poor predictor of proteins' affinity. But the observation that the human and dog states are different is not at all surprising - all it means is that there are multiple amino acid states per residue that are compatible will affinity higher than that conferred by the mouse states. It is very common in multiple sequence alignments to observe some sequence variability at functionally important sites in a protein, such as exchanges between hydrophobic states in a protein's core, or between polar states on a protein's surface, or between donor states (or acceptors) in hydrogen-bonding residues. No selective or epistatic explanation is required to account for this variation - some sequence degeneracy of the functional property is all that is required. The authors' results do show that a search for strict conservation of a single state between proteins with similar affinity is not a reliable guide to identifying sequence sites that contribute to that phenotype, but it would be very naïve to think that it would be. A further consideration related to sequence variability at these sites is that affinity for SARS-Cov2 could not possibly have been a source of long-term constraints that affect sequence variation among species, because the virus did not emerge until 2019. There is no reason that we should expect sites that contribute to affinity to be strictly conserved over evolutionary time.

9. The authors refer to a "functional convergence" between dog and human in their shared susceptibility to SARS-Cov2. But the paper suggests that susceptibility and high ACE2 affinity is ancestral, with a reduction in these phenotypes in the lineage leading to mouse. Susceptibility is therefore not convergent but a retained ancestral state.

I am sorry to say that when these issues are all considered, many of the paper's claims turn out not to be sufficiently supported by the evidence. The paper does establish that six states in mouse can contribute to reducing both affinity and peptidase activity when introduced into human ACE2; further, these states reduce affinity but not activity when introduced into dog ACE2. That is interesting from a sequence-function perspective and should be of interest to scientists whose studies are focused in detail on ACE2 binding and catalysis. But the current paper does not establish whether those states are sufficient to confer resistance in mouse, nor they establish why those states evolved in any of the taxa of interest. Further, there is virtually no evidence for epistasis or an effect on evolutionary processes. The idea of pleiotropic constraints contributing to susceptibility in many lineages is plausible, but the evidence presented does not support it. I do appreciate this paper's effort to connect genetic experiments in ACE2 to evolutionary processes, but the paper's claims on this subject are not justified by the evidence. In my opinion, then, the work reported here should be reported to specialists in the field, but in doing so the claims should be dramatically narrowed.

---

## [Decision Letter · Decision Letter 3]

29 Nov 2021

Dear Dr Duh,

Thank you for submitting your revised Research Article entitled "Evolutionary pathways to SARS-CoV-2 immunity are opened and closed by epistasis" for publication in PLOS Biology. I have now obtained advice from one of the original reviewers and have discussed their comments with the Academic Editor. 

Based on the reviews, we will probably accept this manuscript for publication, provided you satisfactorily address the remaining points raised by the reviewers. Please also make sure to address the following data and other policy-related requests.

IMPORTANT:

a) We agree with reviewer #1 regarding the use of the word "immunity," which we feel is potentially misleading for the reasons that the reviewer identifies. This should be changed throughout, including in the Title. We would favour the terms "resistance" or "lack of susceptibility." Thus the Title should be changed to "Evolutionary pathways to SARS-CoV-2 resistance are opened and closed by epistasis." Additionally it might be helpful to mention ACE2 in the title; perhaps "Evolutionary pathways to SARS-CoV-2 resistance are opened and closed by epistasis acting on ACE2"?

b) We also think that the doom-laden fatalism should be avoided where it occurs - one example is the last sentence of the Abstract, where "..opened the road to immunity for some species, while dooming humans to SARS-CoV-2 susceptibility millions of years before the pandemic started" should perhaps be changed to something like "...opened the road to resistance for some species, while making humans susceptible to viruses that use these ACE2 surfaces for binding, as does SARS-CoV-2."

c) We think that you paper, which is succinct, and has a relatively straightforward message, would be better published as a Short Report. This does not have any formatting implications, but please can you select "Short Reports" as the article type when you resubmit?

d) Please attend to all of the remaining points raised by reviewer #1.

e) Please address my Data Policy requests below; specifically, we need you to supply the numerical values underlying Figs 1ABCDEFG, 2CDEFGHIJKLM, 3CDEF, 4BDEFG, S1, and tree files for 4A, S4, S6, S7. Please cite the location of the data clearly in each relevant main and supplementary Fig legend.

We expect to receive your revised manuscript within two weeks. 

*Published Peer Review History*

*Early Version*

Sincerely,

Roli Roberts

Senior Editor,

rroberts@plos.org,

PLOS Biology

DATA POLICY:

Regardless of the method selected, please ensure that you provide the individual numerical values that underlie the summary data displayed in the following figure panels as they are essential for readers to assess your analysis and to reproduce it: Figs 1ABCDEFG, 2CDEFGHIJKLM, 3CDEF, 4BDEFG, S1; also tree files for 4A, S4, S6, S7. NOTE: the numerical data provided should include all replicates AND the way in which the plotted mean and errors were derived (it should not present only the mean/average values).

We require the original, uncropped and minimally adjusted images supporting all blot and gel results reported in an article's figures or Supporting Information files. We will require these files before a manuscript can be accepted so please prepare and upload them now. Please carefully read our guidelines for how to prepare and upload this data: https://journals.plos.org/plosbiology/s/figures#loc-blot-and-gel-reporting-requirements 

DATA NOT SHOWN?

REVIEWER'S COMMENTS:

Reviewer #1:

The authors have performed considerable work to streamline the manuscript in light of reviewer comments. I still have some various nitpicky details, of minor to modest importance, as I still feel there are some errant claims and comments in the manuscript. In addition to the specific calls below, I feel as a general tonal comment, speculations made beyond conclusions across the manuscript stray a bit far and could be reined in for a tighter interpretation. However, I am better able to appreciate the logical flow of this manuscript than in its prior form. My additional comments are below - all should be achievable via changes in writing with no further experiments requested from my reading. Comments are in order of reading, not importance.

1. Most dramatically in the title and abstract, but also used throughout the manuscript: I dislike the use of "immunity" to describe resistance to SARS-CoV-2 infection in ACE2 variants that resist binding by SARS-CoV-2. In the virology and immunology field, the suggestion of "immunity" would immediately invoke something about innate or adaptive immune system conferring "immunity" to infection by the virus. I believe instead of "immunity", phrasing about "susceptibility' versus perhaps "resistance" (though there may be a better word still than resistance) would be much preferred over "immunity"

2. Lines 43-46: RaTG13 is from R. affinis, not R. sinicus. There is no supported role for recombination with a pangolin virus nor support for pangolin as an intermediate source at this time. To establish zoonotic origin, should be fine to just leave it as referencing circulation of related viruses in several Rhinolophus bat species (at this point, can cite other sarbecoviruses more closely related than RaTG13 as well, and probably don't need to name specific isolates as much as generally point out closely related bat viruses). If needing to discuss why pangolin is included, I would summarize pangolins at this point as another species susceptible to SARS-CoV-2-related CoV infection and spillover, though the routes of exposure and transmission in pangolin communities (e.g. natural circulation, animal trafficking, etc.) are at present unclear.

3. Line 64-67: interaction interface of SARS1 and SARS2 is homologous, and this dichotomy between SARS1 susceptibility to a single mutation and SARS2 targeting multiple hotspots is not a supported claim to my knowledge. SARS1 structure papers also discuss "hotspot" ACE2 residues for interaction, and citations 24-26 don't establish that the single mutation in citation 23 that reduces SARS1 binding does not also reduce SARS2 binding. This differentiation between SARS1 and SARS2 structural mechanisms of ACE2 binding is not well supported and is not necessary to the paper.

4. Line 72-74: somehow missing in this section is consideration that for most species, apart from natural bat reservoirs, there might not be selective pressure to evade binding by SARS-related coronaviruses because these viruses do not naturally circulate in these other species. The current description around antagonistic pleiotropy seems to assume that there naturally would be selective pressure in all of these species to evade binding by SARS-related coronaviruses, thereby requiring some additional explanation, but there need not be any reason in species beyond bats why resistance has evolved that is proximally related to coronavirus susceptibility 

5. Line 126 and on: Shortening of SARS-CoV-2 and -1 to simply "CoV-2" is not standard. Ok in the figure legends for conciseness, but I would avoid it in the written text.

6. Fig. 1: the first time pangolin's and what I have to assume is a bat's silhouettes appear, there is no written label to clarify what ACE2 is being shown. Silhouettes are not sufficient

7. Line 238-240: result seems overstated - K353H does still have a noticeable effect on RBD association in the flow-based assay. Given likely nonlinear relationship between true RBD:ACE2 affinity and labeling in this flow-based assay, this difference in complete versus partial abolishment by K353H could be more about starting affinity of SARS-CoV-1 versus SARS-CoV-2 more than an underlying difference of sensitivities of these viral RBDs to this mutation. True affinity measurements would be needed to support this claim.

8. What error bars represent, the number of replicates, and whether they represent technical versus biological replicates should be clarified in all figure legends

9. The orange circle in the Fig. 4B cartoon is indicative of the primate-rodent ancestor (lateral movement is irrelevant on the tree, so it is equivalent in position to the node of the human/mouse common ancestor). Assuming the black wedge is assumed to represent proliferation of rodents, the orange circle should be moved up to the apex of this black wedge.

10. In the paragraph starting 357 - this hypothesis doesn't "explain" why these six substitutions occurred, because the data suggests that although they maintain the higher catalytic rate, they don't further improve it - is that correct? If so, this paragraph currently is written as though explaining why these mutations occurred in the mouse lineage but not the dog - but in either case, as shown in Fig. 4F, in both backgrounds the six mutations have no net change on the base activity - so this is insufficient to explain their occurrence in mouse but not in dog which in my current reading is the 'purpose' of this paragraph

11. In places such as line 421, the assumption is that the mouse combination is the only route to resistance to SARS2 binding. This is assuredly not the case, as bat ACE2 alleles that abolish SARS-CoV-2 binding use different ACE2 substitutions. Therefore, words such as "required" are incorrect (implying necessity), but rather language here and elsewhere should indicate the mouse solution is one "sufficient" way to restrict binding. I now see this is caveated as a Discussion point, but that doesn't offset the need to use appropriately cautious language when discussing the results.

---

## [Editor Report · Decision Letter 4]

8 Dec 2021

Dear Dr Duh,

On behalf of my colleagues and the Academic Editor, Andreas Hejnol, I'm pleased to say that we can in principle accept your Short Report "Evolutionary pathways to SARS-CoV-2 resistance are opened and closed by epistasis acting on ACE2" for publication in PLOS Biology, provided you address any remaining formatting and reporting issues. These will be detailed in an email that will follow this letter and that you will usually receive within 2-3 business days, during which time no action is required from you. Please note that we will not be able to formally accept your manuscript and schedule it for publication until you have any requested changes.

PRESS: We frequently collaborate with press offices. If your institution or institutions have a press office, please notify them about your upcoming paper at this point, to enable them to help maximise its impact. If the press office is planning to promote your findings, we would be grateful if they could coordinate with biologypress@plos.org. If you have not yet opted out of the early version process, we ask that you notify us immediately of any press plans so that we may do so on your behalf.

Sincerely, 

Roli Roberts

Roland G Roberts, PhD 

Senior Editor 

PLOS Biology

rroberts@plos.org